# Efficient Variational Inference for Sparse Deep Learning with Theoretical Guarantee

**Jincheng Bai**
Department of Statistics
Purdue University
West Lafayette, IN 47906
`bai45@purdue.edu`

**Qifan Song**
Department of Statistics
Purdue University
West Lafayette, IN 47906
`qfsong@purdue.edu`

**Guang Cheng**
Department of Statistics
Purdue University
West Lafayette, IN 47906
`chengg@purdue.edu`

## Abstract

Sparse deep learning aims to address the challenge of huge storage consumption by deep neural networks, and to recover the sparse structure of target functions. Although tremendous empirical successes have been achieved, most sparse deep learning algorithms are lacking of theoretical support. On the other hand, another line of works have proposed theoretical frameworks that are computationally infeasible. In this paper, we train sparse deep neural networks with a fully Bayesian treatment under spike-and-slab priors, and develop a set of computationally efficient variational inferences via continuous relaxation of Bernoulli distribution. The variational posterior contraction rate is provided, which justifies the consistency of the proposed variational Bayes method. Notably, our empirical results demonstrate that this variational procedure provides uncertainty quantification in terms of Bayesian predictive distribution and is also capable to accomplish consistent variable selection by training a sparse multi-layer neural network.

## 1 Introduction

Dense neural network (DNN) may face various problems despite its huge successes in AI fields. Larger training sets and more complicated network structures improve accuracy in deep learning, but always incur huge storage and computation burdens. For example, small portable devices may have limited resources such as several megabyte memory, while a dense neural networks like ResNet-50 with 50 convolutional layers would need more than 95 megabytes of memory for storage and numerous floating number computation (Cheng et al., 2018). It is therefore necessary to compress deep learning models before deploying them on these hardware limited devices.

In addition, sparse neural networks may recover the potential sparsity structure of the target function, e.g., sparse teacher network in the teacher-student framework (Goldt et al., 2019; Tian, 2018). Another example is from nonparametric regression with sparse target functions, i.e., only a portion of input variables are relevant to the response variable. A sparse network may serve the goal of variable selection (Feng and Simon, 2017; Liang et al., 2018; Ye and Sun, 2018), and is also known to be robust to adversarial samples against $l_\infty$ and $l_2$ attacks (Guo et al., 2018).

Bayesian neural network (BNN), which dates back to MacKay (1992); Neal (1992), comparing with frequentist DNN, possesses the advantages of robust prediction via model averaging and automatic uncertainty quantification (Blundell et al., 2015). Conceptually, BNN can easily induce sparse network selection by assigning discrete prior over all possible network structures. However, challenges remain for sparse BNN including inefficient Bayesian computing issue and the lack of theoretical justification. This work aims to resolve these two important bottlenecks simultaneously, by utilizing variational inference approach (Jordan et al., 1999; Blei et al., 2017). On the computational side, it can reduce the ultra-high dimensional sampling problem of Bayesian computing, to an optimization task which can still be solved by a back-propagation algorithm. On the theoretical side, we provide a proper prior specification, under which the variational posterior distribution converges towards the truth. To the best of our knowledge, this work is the first one that provides a complete package of both theory and computation for sparse Bayesian DNN.

**Related work.** A plethora of methods on sparsifying or compressing neural networks have been proposed (Cheng et al., 2018; Gale et al., 2019). The majority of these methods are pruning-based (Han et al., 2016; Zhu and Gupta, 2018; Frankle and Carbin, 2018), which are ad-hoc on choosing the threshold of pruning and usually require additional training and fine tuning. Some other methods could achieve sparsity during training. For example, Louizos et al. (2018) introduced $l_0$ regularized learning and Mocanu et al. (2018) proposed sparse evolutionary training. However, the theoretical guarantee and the optimal choice of hyperparameters for these methods are unclear. As a more natural solution to enforce sparsity in DNN, Bayesian sparse neural network has been proposed by placing prior distributions on network weights: Blundell et al. (2015) and Deng et al. (2019) considered spike-and-slab priors with a Gaussian and Laplacian spike respectively; Log-uniform prior was used in Molchanov et al. (2017); Ghosh et al. (2018) chose to use the popular horseshoe shrinkage prior. These existing works actually yield posteriors over the dense DNN model space despite applying sparsity induced priors. In order to derive explicit sparse inference results, users have to additionally determine certain pruning rules on the posterior. On the other hand, theoretical works regarding sparse deep learning have been studied in Schmidt-Hieber (2017), Polson and Rockova (2018) and Chérief-Abdellatif (2020), but finding an efficient implementation to close the gap between theory and practice remains a challenge for these mentioned methods.

**Detailed contributions.** We impose a spike-and-slab prior on all the edges (weights and biases) of a neural network, where the spike component and slab component represent whether the corresponding edge is inactive or active, respectively. Our work distinguished itself from prior works on Bayesian sparse neural network by imposing the spike-and-slab prior with the Dirac spike function. Hence automatically, all posterior samples are from exact sparse DNN models.

More importantly, with carefully chosen hyperparameter values, especially the prior probability that each edge is active, we establish the variational posterior consistency, and the corresponding convergence rate strikes the balance of statistical estimation error, variational error and the approximation error. The theoretical results are validated by various simulations and real applications. Empirically we also demonstrate that the proposed method possesses good performance of variable selection and uncertainty quantification. While Feng and Simon (2017); Liang et al. (2018); Ye and Sun (2018) only considered the neural network with single hidden layer for variable selection, we observe correct support recovery for neural networks with multi-layer networks.

## 2 Preliminaries

**Nonparametric regression.** Consider a nonparametric regression model with random covariates

$$Y_i = f_0(X_i) + \epsilon_i,\ i = 1, \ldots, n, \tag{1}$$

where $X_i = (x_{i1}, \ldots, x_{ip})^t \sim \mathcal{U}([-1, 1]^p)$[1], $\mathcal{U}$ denotes the uniform distribution, $\epsilon_i \overset{iid}{\sim} \mathcal{N}(0, \sigma_\epsilon^2)$ is the noise term, and $f_0 : [-1, 1]^p \to \mathbb{R}$ is an underlying true function. For simplicity of analysis, we assume $\sigma_\epsilon$ is known. Denote $D_i = (X_i, Y_i)$ and $D = (D_1, \ldots, D_n)$ as the observations. Let $P_0$ denote the underlying probability measure of data, and $p_0$ denote the corresponding density function.

**Deep neural network.** An $L$-hidden-layer neural network is used to model the data. The number of neurons in each hidden layer is denoted by $p_i$ for $i = 1, \ldots, L$. The weight matrix and bias parameter in each layer are denoted by $W_i \in \mathbb{R}^{p_{i-1} \times p_i}$ and $b_i \in \mathbb{R}^{p_i}$ for $i = 1, \ldots, L + 1$. Let $\sigma(x)$ be the activation function, and for any $r \in \mathbb{Z}^+$ and any $b \in \mathbb{R}^r$, we define $\sigma_b : \mathbb{R}^r \to \mathbb{R}^r$ as $\sigma_b(y_j) = \sigma(y_j - b_j)$ for $j = 1, \ldots, r$. Then, given parameters $\boldsymbol{p} = (p_1, \ldots, p_L)'$ and $\theta = \{W_1, b_1, \ldots, W_L, b_L, W_{L+1}, b_{L+1}\}$, the output of this DNN model can be written as

$$f_\theta(X) = W_{L+1}\sigma_{b_L}(W_L \sigma_{b_{L-1}} \ldots \sigma_{b_1}(W_1 X)) + b_{L+1}. \tag{2}$$

In what follows, with slight abuse of notation, $\theta$ is also viewed as a vector that contains all the coefficients in $W_i$'s and $b_i$'s, , i.e., $\theta = (\theta_1, \ldots, \theta_T)'$, where the length $T := \sum_{l=1}^{L-1} p_{l+1}(p_l + 1) + p_1(p + 1) + (p_L + 1)$.

**Variational inference.** Bayesian procedure makes statistical inferences from the posterior distribution $\pi(\theta|D) \propto \pi(\theta) p_\theta(D)$, where $\pi(\theta)$ is the prior distribution. Since MCMC doesn't scale well on complex Bayesian learning tasks with large datasets, variational inference (Jordan et al., 1999; Blei et al., 2017) has become a popular alternative. Given a variational family of distributions, denoted by $\mathcal{Q}$ [2], it seeks to approximate the true posterior distribution by finding a closest member of $\mathcal{Q}$ in terms of KL divergence:

$$\widehat{q}(\theta) = \underset{q(\theta) \in \mathcal{Q}}{\operatorname{argmin}} \operatorname{KL}(q(\theta)||\pi(\theta|D)). \tag{3}$$

The optimization (3) is equivalent to minimize the negative ELBO, which is defined as

$$\Omega = -\mathbb{E}_{q(\theta)}[\log p_\theta(D)] + \operatorname{KL}(q(\theta)||\pi(\theta)), \tag{4}$$

where the first term in (4) can be viewed as the reconstruction error (Kingma and Welling, 2014) and the second term serves as regularization. Hence, the variational inference procedure minimizes the reconstruction error while being penalized against prior distribution in the sense of KL divergence.

When the variational family is indexed by some hyperparameter $\omega$, i.e., any $q \in \mathcal{Q}$ can be written as $q_\omega(\theta)$, then the negative ELBO is a function of $\omega$ as $\Omega(\omega)$. The KL divergence term in (4) could usually be integrated analytically, while the reconstruction error requires Monte Carlo estimation. Therefore, the optimization of $\Omega(\omega)$ can utilize the stochastic gradient approach (Kingma and Welling, 2014). To be concrete, if all distributions in $\mathcal{Q}$ can be reparameterized as $q_\omega \overset{d}{=} g(\omega, \nu)$[3] for some differentiable function $g$ and random variable $\nu$, then the stochastic estimator of $\Omega(\omega)$ and its gradient are

$$\widetilde{\Omega}^m(\omega) = -\frac{n}{m}\frac{1}{K}\sum_{i=1}^{m}\sum_{k=1}^{K} \log p_{g(\omega, \nu_k)}(D_i) + \operatorname{KL}(q_\omega(\theta)||\pi(\theta))$$

$$\nabla_\omega \widetilde{\Omega}^m(\omega) = -\frac{n}{m}\frac{1}{K}\sum_{i=1}^{m}\sum_{k=1}^{K} \nabla_\omega \log p_{g(\omega, \nu_k)}(D_i) + \nabla_\omega \operatorname{KL}(q_\omega(\theta)||\pi(\theta)), \tag{5}$$

where $D_i$'s are randomly sampled data points and $\nu_k$'s are iid copies of $\nu$. Here, $m$ and $K$ are minibatch size and Monte Carlo sample size, respectively.

## 3 Sparse Bayesian deep learning with spike-and-slab prior

We aim to approximate $f_0$ in the generative model (1) by a sparse neural network. Specifically, given a network structure, i.e. the depth $L$ and the width $\boldsymbol{p}$, $f_0$ is approximated by DNN models $f_\theta$ with sparse parameter vector $\theta \in \Theta = \mathbb{R}^T$. From a Bayesian perspective, we impose a spike-and-slab prior (George and McCulloch, 1993; Ishwaran and Rao, 2005) on $\theta$ to model sparse DNN.

A spike-and-slab distribution is a mixture of two components: a Dirac spike concentrated at zero and a flat slab distribution. Denote $\delta_0$ as the Dirac at 0 and $\gamma = (\gamma_1, \ldots, \gamma_T)$ as a binary vector indicating the inclusion of each edge in the network. The prior distribution $\pi(\theta)$ thus follows:

$$\theta_i|\gamma_i \sim \gamma_i \mathcal{N}(0, \sigma_0^2) + (1 - \gamma_i)\delta_0, \quad \gamma_i \sim \operatorname{Bern}(\lambda), \tag{6}$$

for $i = 1, \ldots, T$, where $\lambda$ and $\sigma_0^2$ are hyperparameters representing the prior inclusion probability and the prior Gaussian variance, respectively. The choice of $\sigma_0^2$ and $\lambda$ play an important role in sparse Bayesian learning, and in Section 4, we will establish theoretical guarantees for the variational inference procedure under proper deterministic choices of $\sigma_0^2$ and $\lambda$. Alternatively, hyperparameters may be chosen via an Empirical Bayesian (EB) procedure, but it is beyond the scope of this work. We assume $\mathcal{Q}$ is in the same family of spike-and-slab laws:

$$\theta_i | \gamma_i \sim \gamma_i \mathcal{N}(\mu_i, \sigma_i^2) + (1 - \gamma_i)\delta_0, \quad \gamma_i \sim \text{Bern}(\phi_i) \tag{7}$$

for $i = 1, \ldots, T$, where $0 \leqslant \phi_i \leqslant 1$.

Comparing to pruning approaches (e.g. Zhu and Gupta, 2018; Frankle and Carbin, 2018; Molchanov et al., 2017) that don't pursue sparsity among bias parameter $b_i$'s, the Bayesian modeling induces posterior sparsity for both weight and bias parameters.

In the literature, Polson and Rockova (2018); Chérief-Abdellatif (2020) imposed sparsity specification as follows $\Theta(L, \boldsymbol{p}, s) = \{\theta \text{ as in model (2)} : ||\theta||_0 \leqslant s\}$ that not only posts great computational challenges, but also requires tuning for optimal sparsity level $s$. For example, Chérief-Abdellatif (2020) shows that given $s$, two error terms occur in the variation DNN inference: 1) the variational error $r_n(L, \boldsymbol{p}, s)$ caused by the variational Bayes approximation to the true posterior distribution and 2) the approximation error $\xi_n(L, \boldsymbol{p}, s)$ between $f_0$ and the best bounded-weight $s$-sparsity DNN approximation of $f_0$. Both error terms $r_n$ and $\xi_n$ depend on $s$ (and their specific forms are given in next section). Generally speaking, as the model capacity (i.e., $s$) increases, $r_n$ will increase and $\xi_n$ will decrease. Hence the optimal choice $s^*$ that strikes the balance between these two is

$$s^* = \underset{s}{\text{argmin}} \{r_n(L, \boldsymbol{p}, s) + \xi_n(L, \boldsymbol{p}, s)\}.$$

Therefore, one needs to develop a selection criteria for $\hat{s}$ such that $\hat{s} \approx s^*$. In contrast, our modeling directly works on the whole sparsity regime without pre-specifying $s$, and is shown later to be capable of automatically attaining the same rate of convergence as if the optimal $s^*$ were known.

# 4   Theoretical results

In this section, we will establish the contraction rate of the variational sparse DNN procedure, without knowing $s^*$. For simplicity, we only consider equal-width neural network (similar as Polson and Rockova (2018)).

The following assumptions are imposed:

**Condition 4.1.** $p_i \equiv N \in \mathbb{Z}^+$ *that can depend on n, and* $\lim T = \infty$.

**Condition 4.2.** $\sigma(x)$ *is 1-Lipschitz continuous.*

**Condition 4.3.** *The hyperparameter* $\sigma_0^2$ *is set to be some constant, and* $\lambda$ *satisfies* $\log(1/\lambda) = O\{(L+1)\log N + \log(p\sqrt{n/s^*})\}$ *and* $\log(1/(1-\lambda)) = O((s^*/T)\{(L+1)\log N + \log(p\sqrt{n/s^*})\})$.

Condition 4.2 is very mild, and includes ReLU, sigmoid and tanh. Note that Condition 4.3 gives a wide range choice of $\lambda$, even including the choice of $\lambda$ independent of $s^*$ (See Theorem 4.1 below).

We first state a lemma on an upper bound for the negative ELBO. Denote the log-likelihood ratio between $p_0$ and $p_\theta$ as $l_n(P_0, P_\theta) = \log(p_0(D)/p_\theta(D)) = \sum_{i=1}^n \log(p_0(D_i)/p_\theta(D_i))$. Given some constant $B > 0$, we define

$$
\begin{aligned}
r_n^* &:= r_n(L, N, s^*) = ((L+1)s^*/n)\log N + (s^*/n)\log(p\sqrt{n/s^*}), \\
\xi_n^* &:= \xi_n(L, N, s^*) = \underset{\theta \in \Theta(L, \boldsymbol{p}, s^*), ||\theta||_\infty \leqslant B}{\inf} ||f_\theta - f_0||_\infty^2.
\end{aligned}
$$

Recall that $r_n(L, N, s)$ and $\xi_n(L, N, s)$ denote the variational error and the approximation error.

**Lemma 4.1.** *Under Condition 4.1-4.3, then with dominating probability,*

$$\underset{q(\theta) \in \mathcal{Q}}{\inf} \left\{ KL(q(\theta)||\pi(\theta|\lambda)) + \int_\Theta l_n(P_0, P_\theta)q(\theta)d\theta \right\} \leqslant Cn(r_n^* + \xi_n^*) \tag{8}$$

*where C is either some positive constant if* $\lim n(r_n^* + \xi_n^*) = \infty$, *or any diverging sequence if* $\limsup n(r_n^* + \xi_n^*) \neq \infty$.

Noting that $\mathrm{KL}(q(\theta)||\pi(\theta|\lambda)) + \int_\Theta l_n(P_0, P_\theta)q(\theta)(d\theta)$ is the negative ELBO up to a constant, we therefore show the optimal loss function of the proposed variational inference is bounded.

The next lemma investigates the convergence of the variational distribution under the Hellinger distance, which is defined as

$$d^2(P_\theta, P_0) = \mathbb{E}_X\left(1 - \exp\{-[f_\theta(X) - f_0(X)]^2/(8\sigma_\epsilon^2)\}\right).$$

In addition, let $s_n = s^* \log^{2\delta-1}(n)$ for any $\delta > 1$. An assumption on $s^*$ is required to strike the balance between $r_n^*$ and $\xi^*$:

**Condition 4.4.** $\max\{s^* \log(p\sqrt{n/s^*}, (L+1)s^* \log N\} = o(n)$ and $r_n^* \asymp \xi_n^*$.

**Lemma 4.2.** *Under Conditions 4.1-4.4, if $\sigma_0^2$ is set to be constant and $\lambda \leqslant T^{-1}\exp\{-Mnr_n^*/s_n\}$ for any positive diverging sequence $M \to \infty$, then with dominating probability, we have*

$$\int_\Theta d^2(P_\theta, P_0)\widehat{q}(\theta)d\theta \leqslant C\varepsilon_n^{*2} + \frac{3}{n}\inf_{q(\theta)\in\mathcal{Q}}\left\{KL(q(\theta)||\pi(\theta|\lambda)) + \int_\Theta l_n(P_0, P_\theta)q(\theta)d\theta\right\}, \quad (9)$$

*where $C$ is some constant, and*

$$\varepsilon_n^* := \varepsilon_n(L, N, s^*) = \sqrt{r_n(L, N, s^*)}\log^\delta(n), \text{ for any } \delta > 1.$$

**Remark.** *The result (9) is of exactly the same form as in the existing literature (Pati et al., 2018), but it is not trivial in the following sense. The existing literature require the existence of a global testing function that separates $P_0$ and $\{P_\theta : d(P_\theta, P_0) \geqslant \varepsilon_n^*\}$ with exponentially decay rate of Type I and Type II errors. If such a testing function exists only over a subset $\Theta_n \subset \Theta$ (which is the case for our DNN modeling), then the existing result (Yang et al., 2020) can only characterize the VB posterior contraction behavior within $\Theta_n$, but not over the whole parameter space $\Theta$. Therefore our result, which characterizes the convergence behavior for the overall VB posterior, represents a significant improvement beyond those works.*

The above two lemmas together imply the following guarantee for VB posterior:

**Theorem 4.1.** *Let $\sigma_0^2$ be a constant and $-\log\lambda = \log(T) + \delta[(L+1)\log N + \log\sqrt{n}p]$ for any constant $\delta > 0$. Under Conditions 4.1-4.2, 4.4, we have with high probability*

$$\int_\Theta d^2(P_\theta, P_0)\widehat{q}(\theta)d\theta \leqslant C\varepsilon_n^{*2} + C'(r_n^* + \xi_n^*),$$

*where $C$ is some positive constant and $C'$ is any diverging sequence.*

The $\varepsilon_n^{*2}$ denotes the estimation error from the statistical estimator for $P_0$. The variational Bayes convergence rate consists of estimating error, i.e., $\varepsilon_n^{*2}$, variational error, i.e., $r_n^*$, and approximation error, i.e., $\xi_n^*$. Given that the former two errors have only logarithmic difference, our convergence rate actually strikes the balance among all three error terms. The derived convergence rate has an explicit expression in terms of the network structure based on the forms of $\varepsilon_n^*$, $r_n^*$ and $\xi_n^*$, in contrast with general convergence results in Pati et al. (2018); Zhang and Gao (2019); Yang et al. (2020).

**Remark.** *Theorem 4.1 provides a specific choice for $\lambda$, which can be relaxed to the general conditions on $\lambda$ in Lemma 4.2. In contrast to the heuristic choices such as $\lambda = \exp(-2\log n)$ (BIC; Hubin and Storvik, 2019), this theoretically justified choice incorporates knowledge of input dimension, network structure and sample size. Such an $\lambda$ will be used in our numerical experiments in Section 6, but readers shall be aware of that its theoretical validity is only justified in an asymptotic sense.*

**Remark.** *The convergence rate is derived under Hellinger metric, which is of less practical relevance than $L_2$ norm representing the common prediction error. One may obtain a convergence result under $L_2$ norm via a VB truncation (refer to supplementary material, Theorem 1.1).*

**Remark.** *If $f_0$ is an $\alpha$-Hölder-smooth function with fixed input dimension $p$, then by choosing some $L \asymp \log n$, $N \asymp n/\log n$, combining with the approximation result (Schmidt-Hieber, 2017, Theorem 3), our theorem ensures rate-minimax convergence up to a logarithmic term.*

## 5 Implementation

To conduct optimization of (4) via stochastic gradient optimization, we need to find certain reparameterization for any distribution in $\mathcal{Q}$. One solution is to use the inverse CDF sampling technique.

Specifically, if $\theta \sim q \in \mathcal{Q}$, its marginal $\theta_i$'s are independent mixture of (7). Let $F_{(\mu_i, \sigma_i, \phi_i)}$ be the CDF of $\theta_i$, then $\theta_i \overset{d}{=} F_{(\mu_i, \sigma_i, \phi_i)}^{-1}(u_i)$ holds where $u_i \sim \mathcal{U}(0, 1)$. This inverse CDF reparameterization, although valid, can not be conveniently implemented within the state-of-art python packages like PyTorch. Rather, a more popular way in VB is to utilize the Gumbel-softmax approximation.

We rewrite the loss function $\Omega$ as

$$-\mathbb{E}_{q(\theta|\gamma)q(\gamma)}[\log p_\theta(D)] + \sum_{i=1}^{T} \text{KL}(q(\gamma_i)||\pi(\gamma_i)) + \sum_{i=1}^{T} q(\gamma_i = 1)\text{KL}(\mathcal{N}(\mu_i, \sigma_i^2)||\mathcal{N}(0, \sigma_0^2)). \quad (10)$$

Since it is impossible to reparameterize the discrete variable $\gamma$ by a continuous system, we apply continuous relaxation, i.e., to approximate $\gamma$ by a continuous distribution. In particular, the Gumbel-softmax approximation (Maddison et al., 2017; Jang et al., 2017) is used here, and $\gamma_i \sim \text{Bern}(\phi_i)$ is approximated by $\widetilde{\gamma}_i \sim \text{Gumbel-softmax}(\phi_i, \tau)$, where

$$\widetilde{\gamma}_i = (1 + \exp(-\eta_i/\tau))^{-1}, \quad \eta_i = \log \frac{\phi_i}{1 - \phi_i} + \log \frac{u_i}{1 - u_i}, \quad u_i \sim \mathcal{U}(0, 1).$$

$\tau$ is called the temperature, and as it approaches 0, $\tilde{\gamma}_i$ converges to $\gamma_i$ in distribution (refer to Figure 1 in the supplementary material). In addition, one can show that $P(\widetilde{\gamma}_i > 0.5) = \phi_i$, which implies $\gamma_i \overset{d}{=} 1(\widetilde{\gamma}_i > 0.5)$. Thus, $\widetilde{\gamma}_i$ is viewed as a soft version of $\gamma_i$, and will be used in the backward pass to enable the calculation for gradients, while the hard version $\gamma_i$ will be used in the forward pass to obtain a sparse network structure. In practice, $\tau$ is usually chosen no smaller than 0.5 for numerical stability. Besides, the normal variable $\mathcal{N}(\mu_i, \sigma_i^2)$ is reparameterized by $\mu_i + \sigma_i \epsilon_i$ for $\epsilon_i \sim \mathcal{N}(0, 1)$.

The complete variational inference procedure with Gumbel-softmax approximation is stated below.

---

**Algorithm 1** Variational inference for sparse BNN with normal slab distribution.

---

1: parameters: $\omega = (\mu, \sigma', \phi')$,
2: where $\sigma_i = \log(1 + \exp(\sigma_i'))$, $\phi_i = (1 + \exp(\phi_i'))^{-1}$, for $i = 1, \dots, T$
3: **repeat**
4:     $D^m \leftarrow$ Randomly draw a minibatch of size $m$ from $D$
5:     $\epsilon_i, u_i \leftarrow$ Randomly draw $K$ samples from $\mathcal{N}(0, 1)$ and $\mathcal{U}(0, 1)$
6:     $\widetilde{\Omega}^m(\omega) \leftarrow$ Use (5) with $(D^m, \omega, \epsilon, u)$; Use $\gamma$ in the forward pass
7:     $\nabla_\omega \widetilde{\Omega}^m(\omega) \leftarrow$ Use (5) with $(D^m, \omega, \epsilon, u)$; Use $\widetilde{\gamma}$ in the backward pass
8:     $\omega \leftarrow$ Update with $\nabla_\omega \widetilde{\Omega}^m(\omega)$ using gradient descent algorithms (e.g. SGD or Adam)
9: **until** convergence of $\widetilde{\Omega}^m(\omega)$
10: **return** $\omega$

---

The Gumbel-softmax approximation introduces an additional error that may jeopardize the validity of Theorem 4.1. Our exploratory studies (refer to Section 2.3 in supplementary material) demonstrates little differences between the results of using inverse-CDF reparameterization and using Gumbel-softmax approximation in some simple model. Therefore, we conjecture that Gumbel-softmax approximation doesn't hurt the VB convergence, and thus will be implemented in our numerical studies.

## 6 Experiments

We evaluate the empirical performance of the proposed variational inference through simulation study and MNIST data application. For the simulation study, we consider a teacher-student framework and a nonlinear regression function, by which we justify the consistency of the proposed method and validate the proposed choice of hyperparameters. As a byproduct, the performance of uncertainty quantification and the effectiveness of variable selection will be examined as well.

For all the numerical studies, we let $\sigma_0^2 = 2$, the choice of $\lambda$ follows Theorem 4.1 (denoted by $\lambda_{opt}$): $\log(\lambda_{opt}^{-1}) = \log(T) + 0.1[(L + 1)\log N + \log \sqrt{n}p]$. The remaining details of implementation (such as initialization, choices of $K$, $m$ and learning rate) are provided in the supplementary material. We will use VB posterior mean estimator $\widehat{f}_H = \sum_{h=1}^{H} f_{\theta_h}/H$ to assess the prediction accuracy, where

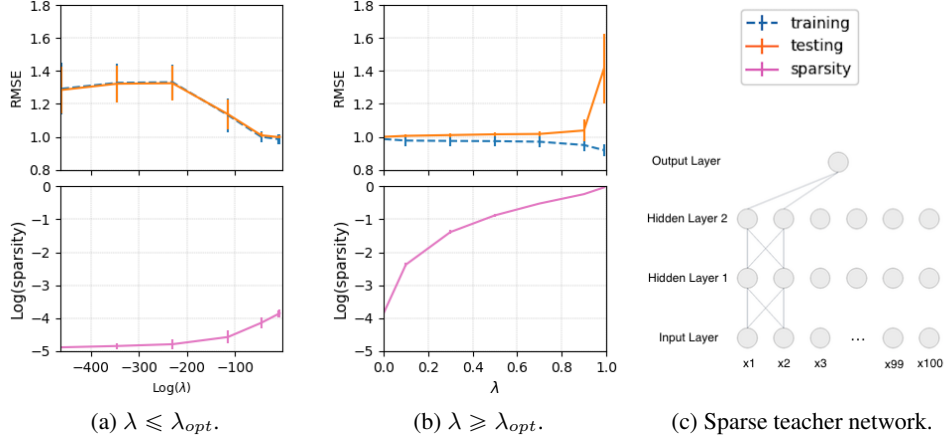

(a) $\lambda \leqslant \lambda_{opt}$.  (b) $\lambda \geqslant \lambda_{opt}$.  (c) Sparse teacher network.

Figure 1: (a) $\lambda = \{10^{-200}, 10^{-150}, 10^{-100}, 10^{-50}, 10^{-20}, 10^{-5}, \lambda_{opt}\}$. (b) $\lambda = \{\lambda_{opt}, 0.1, 0.3, 0.5, 0.7, 0.9, 0.99\}$. (c) The structure of the target sparse teacher network. Please note that the $x$ axes of figures (a) and (b) are in different scales.

$\theta_h \sim \hat{q}(\theta)$ are samples drawn from the VB posterior and $H = 30$. The posterior network sparsity is measured by $\hat{s} = \sum_{i=1}^{T} \phi_i / T$. Input nodes who have connection with $\phi_i > 0.5$ to the second layer is selected as relevant input variables, and we report the corresponding false positive rate (FPR) and false negative rate (FNR) to evaluate the variable selection performance of our method.

Our method will be compared with the dense variational BNN (VBNN) (Blundell et al., 2015) with independent centered normal prior and independent normal variational distribution, the AGP pruner (Zhu and Gupta, 2018), the Lottery Ticket Hypothesis (LOT) (Frankle and Carbin, 2018), the variational dropout (VD) (Molchanov et al., 2017) and the Horseshoe BNN (HS-BNN) (Ghosh et al., 2018). In particular, VBNN can be regarded as a baseline method without any sparsification or compression. All reported simulation results are based on 30 replications (except that we use 60 replications for interval estimation coverages). Note that the sparsity level in methods AGP and LOT are user-specified. Hence, in simulation studies, we try a grid search for AGP and LOT, and only report the ones that yield highest testing accuracy. Furthermore, note that FPR and FNR are not calculated for HS-BNN since it only sparsifies the hidden layers nodewisely.

**Simulation I: Teacher-student networks setup**  We consider two teacher network settings for $f_0$: (A) densely connected with a structure of 20-6-6-1, $p = 20$, $n = 3000$, $\sigma(x) = \text{sigmoid}(x)$, $X \sim \mathcal{U}([-1, 1]^{20})$, $\epsilon \sim \mathcal{N}(0, 1)$ and network parameter $\theta_i$ is randomly sampled from $\mathcal{U}(0, 1)$; (B) sparsely connected as shown in Figure 1 (c), $p = 100$, $n = 500$, $\sigma(x) = \tanh(x)$, $X \sim \mathcal{U}([-1, 1]^{100})$ and $\epsilon \sim \mathcal{N}(0, 1)$, the network parameter $\theta_i$'s are fixed (refer to Section 2.4 in the supplementary material for details).

Table 1: Simulation results for Simulation I. SVBNN represents our sparse variational BNN method. The sparsity levels specified for AGP are 30% and 5%, and for LOT are 10% and 5%, respectively for the two cases.

|  | | RMSE | | Input variable selection | | | |
| --- | --- | --- | --- | --- | --- | --- | --- |
|  | **Method** | **Train** | **Test** | **FPR(%)** | **FNR(%)** | **95% Coverage (%)** | **Sparsity(%)** |
| Dense | SVBNN | $1.01 \pm 0.02$ | $1.01 \pm 0.00$ | - | - | $97.5 \pm 1.71$ | $6.45 \pm 0.83$ |
|  | VBNN | $1.00 \pm 0.02$ | $1.00 \pm 0.00$ | - | - | $91.4 \pm 3.89$ | $100 \pm 0.00$ |
|  | VD | $0.99 \pm 0.02$ | $1.01 \pm 0.00$ | - | - | $76.4 \pm 4.75$ | $28.6 \pm 2.81$ |
|  | HS-BNN | $0.98 \pm 0.02$ | $1.02 \pm 0.01$ | - | - | $83.5 \pm 0.78$ | $64.9 \pm 24.9$ |
|  | AGP | $0.99 \pm 0.02$ | $1.01 \pm 0.00$ | - | - | - | $30.0 \pm 0.00$ |
|  | LOT | $1.04 \pm 0.01$ | $1.02 \pm 0.00$ | - | - | - | $10.0 \pm 0.00$ |
| Sparse | SVBNN | $0.99 \pm 0.03$ | $1.00 \pm 0.01$ | $0.00 \pm 0.00$ | $0.00 \pm 0.00$ | $96.4 \pm 4.73$ | $2.15 \pm 0.25$ |
|  | VBNN | $0.92 \pm 0.05$ | $1.53 \pm 0.17$ | $100 \pm 0.00$ | $0.00 \pm 0.00$ | $90.7 \pm 8.15$ | $100 \pm 0.00$ |
|  | VD | $0.86 \pm 0.04$ | $1.07 \pm 0.03$ | $72.9 \pm 6.99$ | $0.00 \pm 0.00$ | $75.5 \pm 7.81$ | $20.8 \pm 3.08$ |
|  | HS-BNN | $0.90 \pm 0.04$ | $1.29 \pm 0.04$ | - | - | $67.0 \pm 8.54$ | $32.1 \pm 20.1$ |
|  | AGP | $1.01 \pm 0.03$ | $1.02 \pm 0.00$ | $16.9 \pm 1.81$ | $0.00 \pm 0.00$ | - | $5.00 \pm 0.00$ |
|  | LOT | $0.96 \pm 0.01$ | $1.04 \pm 0.01$ | $19.5 \pm 2.57$ | $0.00 \pm 0.00$ | - | $5.00 \pm 0.00$ |

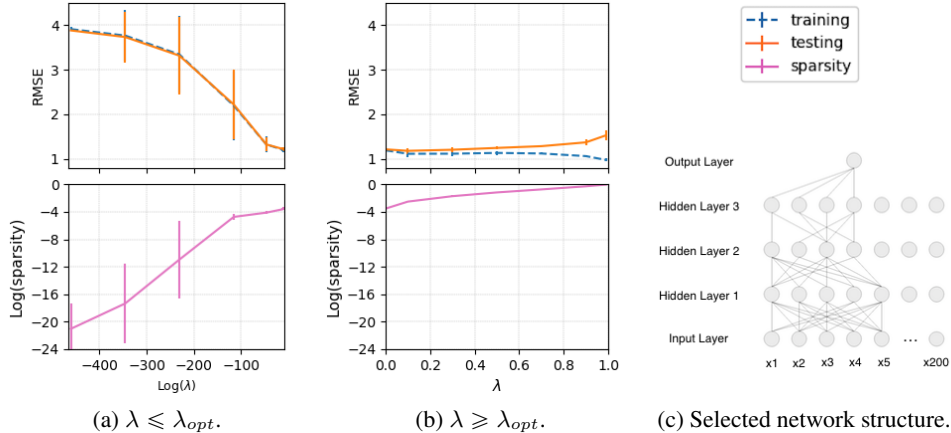

(a) $\lambda \leqslant \lambda_{opt}$.　　　　(b) $\lambda \geqslant \lambda_{opt}$.　　　　(c) Selected network structure.

Figure 2: (a) $\lambda = \{10^{-200}, 10^{-150}, 10^{-100}, 10^{-50}, 10^{-20}, 10^{-5}, \lambda_{opt}\}$. (b) $\lambda = \{\lambda_{opt}, 0.1, 0.3, 0.5,$ $0.7, 0.9, 0.99\}$. (c) A selected network structure for (11).

First, we examine the impact of different choices of $\lambda$ on our VB sparse DNN modeling. A set of different $\lambda$ values are used, and for each $\lambda$, we compute the training square root MSE (RMSE) and testing RMSE based on $\widehat{f}_H$. Results for the simulation setting (B) are plotted in Figure 1 along with error bars (Refer to Section 2.4 in supplementary material for the plot under the simulation setting (A)). The figure shows that as $\lambda$ increases, the resultant network becomes denser and the training RMSE monotonically decreases, while testing RMSE curve is roughly U-shaped. In other words, an overly small $\lambda$ leads to over-sparsified DNNs with insufficient expressive power, and an overly large $\lambda$ leads to overfitting DNNs. The suggested $\lambda_{opt}$ successfully locates in the valley of U-shaped testing curve, which empirically justifies our theoretical choice of $\lambda_{opt}$.

We next compare the performance of our method (with $\lambda_{opt}$) to the benchmark methods, and present results in Table 1. For the dense teacher network (A), our method leads to the most sparse structure with comparable prediction error; For the sparse teacher network (B), our method not only achieves the best prediction accuracy, but also always selects the correct set of relevant input variables. Besides, we also explore uncertainty quantification of our methods, by studying the coverage of 95% Bayesian predictive intervals (refer to supplementary material for details). Table 1 shows that our method obtains coverage rates slightly higher than the nominal levels while other (Bayesian) methods suffer from undercoverage problems.

**Simulation II: Sparse nonlinear function**　　Consider the following sparse function $f_0$:

$$f_0(x_1, \ldots, x_{200}) = \frac{7x_2}{1 + x_1^2} + 5\sin(x_3 x_4) + 2x_5, \quad \epsilon \sim \mathcal{N}(0, 1), \tag{11}$$

all covariates are iid $\mathcal{N}(0, 1)$ and data set contains $n = 3000$ observations. A ReLU network with $L = 3$ and $N = 7$ is used. Similar to the simulation I, we study the impact of $\lambda$, and results in Figure 2 justify that $\lambda_{opt}$ is a reasonable choice. Table 2 compares the performances of our method (under $\lambda_{opt}$) to the competitive methods. Our method exhibits the best prediction power with minimal connectivity, among all the methods. In addition, our method achieves smallest FPR and acceptable FNR for input variable selection. In comparison, other methods select huge number of false input variables. Figure 2 (c) shows the selected network (edges with $\phi_i > 0.5$) in one replication that correctly identifies the input variables.

**MNIST application.**　　We evaluate the performance of our method on MNIST data for classification tasks, by comparing with benchmark methods. A 2-hidden layer DNN with 512 neurons in each layer is used. We compare the testing accuracy of our method (with $\lambda_{opt}$) to the benchmark methods at different epochs using the same batch size (refer to supplementary material for details). Figure 3 shows our method achieves best accuracy as epoch increases, and the final sparsity level for SVBNN, AGP and VD are $5.06\%$, $5.00\%$ and $2.28\%$.

Table 2: Results for Simulation II. The sparsity levels selected for AGP and LOT are both 30%.

| Method | Train RMSE | Test RMSE | FPR(%) | FNR(%) | Sparsity(%) |
|---|---|---|---|---|---|
| SVBNN | $1.19 \pm 0.05$ | $1.21 \pm 0.05$ | $0.00 \pm 0.21$ | $16.0 \pm 8.14$ | $2.97 \pm 0.48$ |
| VBNN | $0.96 \pm 0.06$ | $1.99 \pm 0.49$ | $100 \pm 0.00$ | $0.00 \pm 0.00$ | $100 \pm 0.00$ |
| VD | $1.02 \pm 0.05$ | $1.43 \pm 0.19$ | $98.6 \pm 1.22$ | $0.67 \pm 3.65$ | $46.9 \pm 4.72$ |
| HS-BNN | $1.17 \pm 0.52$ | $1.66 \pm 0.43$ | - | - | $41.1 \pm 36.5$ |
| AGP | $1.06 \pm 0.08$ | $1.58 \pm 0.11$ | $82.7 \pm 3.09$ | $1.33 \pm 5.07$ | $30.0 \pm 0.00$ |
| LOT | $1.08 \pm 0.09$ | $1.44 \pm 0.14$ | $83.6 \pm 2.94$ | $0.00 \pm 0.00$ | $30.0 \pm 0.00$ |

In addition, an illustration of our method's capability for uncertainty quantification on MNIST can be found in the supplementary material, where additional experimental results on UCI regression datasets can also be found.

## 7 Conclusion and discussion

We proposed a variational inference method for deep neural networks under spike-and-slab priors with theoretical guarantees. Future direction could be investigating the theory behind choosing hyperparamters via the EB estimation instead of deterministic choices.

Furthermore, extending the current results to more complicated networks (convolutional neural network, residual network, etc.) is not trivial. Conceptually, it requires the design of structured sparsity (e.g., group sparsity in Neklyudov et al. (2017)) to fulfill the goal of faster prediction. Theoretically, it requires deeper understanding of the expressive ability (i.e. approximation error) and capacity (i.e., packing or covering number) of the network model space. For illustration

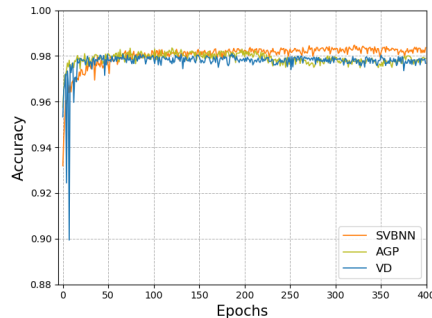

Figure 3: Testing accuracy for MNIST

purpose, we include an example of Fashion-MNIST task using convolutional neural network in the supplementary material, and it demonstrates the usage of our method on more complex networks in practice.

## Broader Impact

We believe the ethical aspects are not applicable to this work. For future societal consequences, deep learning has a wide range of applications such as computer version and natural language processing. Our work provides a solution to overcome the drawbacks of modern deep neural network, and also improves the understanding of deep learning.

The proposed method could improve the existing applications. Specifically, sparse learning helps apply deep neural networks to hardware limited devices, like cell phones or pads, which will broaden the horizon of deep learning application. In addition, as a Bayesian method, not only a result, but also the knowledge of confidence or certainty in that result are provided, which could benefit people in various aspects. For example, in the application of cancer diagnostic, by providing the certainty associated with each possible outcome, Bayesian learning would assist the medical professionals to make a better judgement about whether the tumor is a cancer or a benign one. Such kind of ability to quantify uncertainty would contribute to the modern deep learning.

## Acknowledgments and Disclosure of Funding

We would like to thank Wei Deng for the helpful discussion and thank the reviewers for their thoughtful comments. Qifan Song's research is partially supported by National Science Foundation

grant DMS-1811812. Guang Cheng was a member of the Institute for Advanced Study in writing this paper, and he would like to thank the institute for its hospitality.

## Footnotes

[1]This compact support assumption is generally satisfied given the standardized data, and may be relaxed.

[2]For simplicity, it is commonly assumed that $\mathcal{Q}$ is the mean-field family, i.e. $q(\theta) = \prod_{i=1}^{T} q(\theta_i)$.

[3]"$\overset{d}{=}$" means equivalence in distribution

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
