[Supplementary Material]

# Supplementary Document to the Paper "Efficient Variational Inference for Sparse Deep Learning with Theoretical Guarantee"

**Jincheng Bai**
Department of Statistics
Purdue University
West Lafayette, IN 47906
`bai45@purdue.edu`

**Qifan Song**
Department of Statistics
Purdue University
West Lafayette, IN 47906
`qfsong@purdue.edu`

**Guang Cheng**
Department of Statistics
Purdue University
West Lafayette, IN 47906
`chengg@purdue.edu`

In this document, the detailed proofs for the theoretical results are provided in the first section, along with additional numerical results presented in the second section.

## 1 Proofs of theoretical results

### 1.1 Proof of Lemma 4.1

As a technical tool for the proof, we first restate the Lemma 6.1 in Chérief-Abdellatif and Alquier (2018) as follows.

**Lemma 1.1.** *For any $K > 0$, the KL divergence between any two mixture densities $\sum_{k=1}^{K} w_k g_k$ and $\sum_{k=1}^{K} \tilde{w}_k \tilde{g}_k$ is bounded as*

$$KL(\sum_{k=1}^{K} w_k g_k || \sum_{k=1}^{K} \tilde{w}_k \tilde{g}_k) \leqslant KL(\boldsymbol{w}||\tilde{\boldsymbol{w}}) + \sum_{k=1}^{K} w_k KL(g_k||\tilde{g}_k),$$

*where $KL(\boldsymbol{w}||\tilde{\boldsymbol{w}}) = \sum_{k=1}^{K} w_k \log \frac{w_k}{\tilde{w}_k}$.*

**Proof of Lemma 4.1**

*Proof.* It suffices to construct some $q^*(\theta) \in \mathcal{Q}$, such that w.h.p,

$$\mathrm{KL}(q^*(\theta)||\pi(\theta|\lambda)) + \int_{\Theta} l_n(P_0, P_\theta) q^*(\theta)(d\theta)$$
$$\leqslant C_1 n r_n^* + C_1' n \inf_{\theta} ||f_\theta - f_0||_\infty^2 + C_1' n r_n^*,$$

where $C_1$, $C_1'$ are some positive constants if $\lim n(r_n^* + \xi_n^*) = \infty$, or any diverging sequences if $\limsup n(r_n^* + \xi_n^*) \neq \infty$.

Recall $\theta^* = \arg\min_{\theta \in \Theta(L, \boldsymbol{p}, s^*, B)} ||f_\theta - f_0||_\infty^2$, then $q^*(\theta) \in \mathcal{Q}$ can be constructed as

$$\mathrm{KL}(q^*(\theta)||\pi(\theta|\lambda)) \leqslant C_1 n r_n^*, \tag{1}$$

$$\int_{\Theta} ||f_\theta - f_{\theta*}||_\infty^2 q^*(\theta)(d\theta) \leqslant r_n^*. \tag{2}$$

We define $q^*(\theta)$ as follows, for $i = 1, \ldots, T$:

$$\theta_i | \gamma_i^* \sim \gamma_i^* \mathcal{N}(\theta_i^*, \sigma_n^2) + (1 - \gamma_i^*)\delta_0,$$
$$\gamma_i^* \sim \text{Bern}(\phi_i^*), \tag{3}$$
$$\phi_i^* = 1(\theta_i^* \neq 0),$$

where $a_n^2 = \frac{s^*}{8n} \log^{-1}(3pN)(2BN)^{-2(L+1)} \left\{ (p+1+\frac{1}{BN-1})^2 + \frac{1}{(2BN)^2-1} + \frac{2}{(2BN-1)^2} \right\}^{-1}$.

To prove (1), denote $\Gamma^T$ as the set of all possible binary inclusion vectors with length $T$, then $q^*(\theta)$ and $\pi(\theta|\lambda)$ could be written as mixtures

$$q^*(\theta) = \sum_{\gamma \in \Gamma^T} 1(\gamma = \gamma^*) \prod_{i=1}^{T} \gamma_i \mathcal{N}(\theta_i^*, \sigma_n^2) + (1 - \gamma_i)\delta_0,$$

and

$$\pi(\theta|\lambda) = \sum_{\gamma \in \Gamma^T} \pi(\gamma) \prod_{i=1}^{T} \gamma_i \mathcal{N}(0, \sigma_0^2) + (1 - \gamma_i)\delta_0,$$

where $\pi(\gamma)$ is the probability for vector $\gamma$ under prior distribution $\pi$. Then,

$\text{KL}(q^*(\theta)||\pi(\theta|\lambda))$

$$\leq \log \frac{1}{\pi(\gamma^*)} + \sum_{\gamma \in \Gamma^T} 1(\gamma = \gamma^*)\text{KL}\left\{ \prod_{i=1}^{T} \gamma_i \mathcal{N}(\theta_i^*, \sigma_n^2) + (1 - \gamma_i)\delta_0 \Big\| \prod_{i=1}^{T} \gamma_i \mathcal{N}(0, \sigma_0^2)) + (1 - \gamma_i)\delta_0 \right\}$$

$$= \log \frac{1}{\lambda^{s^*}(1-\lambda)^{T-s^*}} + \sum_{i=1}^{T} \text{KL}\left\{ \gamma_i^* \mathcal{N}(\theta_i^*, \sigma_n^2) + (1 - \gamma_i^*)\delta_0 || \gamma_i^* \mathcal{N}(0, \sigma_0^2)) + (1 - \gamma_i^*)\delta_0 \right\}$$

$$= s^* \log(\frac{1}{\lambda}) + (T - s^*)\log(\frac{1}{1-\lambda}) + \sum_{i=1}^{T} \gamma_i^* \left\{ \frac{1}{2} \log\left(\frac{\sigma_0^2}{\sigma_n^2}\right) + \frac{\sigma_n^2 + \theta_i^{*2}}{2} - \frac{1}{2} \right\}$$

$$\leq C_0 n r_n^* + \frac{s^*}{2}\sigma_n^2 + \frac{s^*}{2}(B^2 - 1) + \frac{s^*}{2}\log\left(\frac{\sigma_0^2}{\sigma_n^2}\right)$$

$$\leq (C_0 + 1)n r_n^* + \frac{s^*}{2}B^2 + \frac{s^*}{2}\log\left(\frac{8n}{s^*}\log(3pN)(2BN)^{2L+2}\left\{(p+1+\frac{1}{BN-1})^2\right.\right.$$

$$\left.\left. + \frac{1}{(2BN)^2-1} + \frac{2}{(2BN-1)^2}\right\}\right)$$

$$\leq (C_0 + 2)n r_n^* + \frac{B^2}{2}s^* + (L+1)s^*\log(2BN) + \frac{s^*}{2}\log\log(3BN) + \frac{s^*}{2}\log\left(\frac{n}{s^*}p^2\right)$$

$$\leq (C_0 + 3)n r_n^* + (L+1)s^*\log N + s^*\log\left(p\sqrt{\frac{n}{s^*}}\right)$$

$$\leq C_1 n r_n^*, \text{ for sufficiently large n,}$$

where $C_0$ and $C_1$ are some fixed constants. The first inequality is due to Lemma 1.1 and the second inequality is due to Condition 4.4.

Furthermore, by Appendix G of Chérief-Abdellatif (2020), it can be shown

$$\int_{\Theta} ||f_\theta - f_{\theta*}||_\infty^2 q^*(\theta)(d\theta)$$

$$\leq 8a_n^2 \log(3BN)(2BN)^{2L+2}\left\{(p+1+\frac{1}{BN-1})^2 + \frac{1}{(2BN)^2-1} + \frac{2}{(2BN-1)^2}\right\}$$

$$\leq \frac{s^*}{n} \leq r_n^*.$$

Noting that

$$l_n(P_0, P_\theta) = \frac{1}{2\sigma_\epsilon^2}(||Y - f_\theta(X)||_2^2 - ||Y - f_0(X)||_2^2)$$

$$= \frac{1}{2\sigma_\epsilon^2}(||Y - f_0(X) + f_0(X) - f_\theta(X)||_2^2 - ||Y - f_0(X)||_2^2)$$

$$= \frac{1}{2\sigma_\epsilon^2}(||f_\theta(X) - f_0(X)||_2^2 + 2\langle Y - f_0(X), f_0(X) - f_\theta(X)\rangle),$$

Denote

$$\mathcal{R}_1 = \int_\Theta ||f_\theta(X) - f_0(X)||_2^2 q^*(\theta)(d\theta),$$

$$\mathcal{R}_2 = \int_\Theta \langle Y - f_0(X), f_0(X) - f_\theta(X)\rangle q^*(\theta)(d\theta).$$

Since $||f_\theta(X) - f_0(X)||_2^2 \leqslant n||f_\theta - f_0||_\infty^2 \leqslant n||f_\theta - f_{\theta*}||_\infty^2 + n||f_{\theta*} - f_0||_\infty^2$,

$$\mathcal{R}_1 \leqslant nr_n^* + n||f_{\theta*} - f_0||_\infty^2.$$

Noting that $Y - f_0(X) = \epsilon \sim \mathcal{N}(0, \sigma_\epsilon^2 I)$, then

$$\mathcal{R}_2 = \int_\Theta \epsilon^T(f_0(X) - f_\theta(X))q^*(\theta)(d\theta) = \epsilon^T \int_\Theta (f_0(X) - f_\theta(X))q^*(\theta)(d\theta) \sim \mathcal{N}(0, c_f\sigma_\epsilon^2),$$

where $c_f = ||\int_\Theta (f_0(X) - f_\theta(X))q^*(\theta)(d\theta)||_2^2 \leqslant \mathcal{R}_1$ due to Cauchy-Schwarz inequality. Therefore, $\mathcal{R}_2 = O_p(\sqrt{\mathcal{R}_1})$, and w.h.p., $\mathcal{R}_2 \leqslant C_0'\mathcal{R}_1$, where $C_0'$ is some positive constant if $\lim n(r_n^* + \xi_n^*) = \infty$ or $C_0'$ is any diverging sequence if $\limsup n(r_n^* + \xi_n^*) \neq \infty$. Therefore,

$$\int_\Theta l_n(P_0, P_\theta)q^*(\theta)(d\theta) = \mathcal{R}_1/2\sigma_\epsilon^2 + \mathcal{R}_2/\sigma_\epsilon^2 \leqslant (2C_0' + 1)n(r_n^* + ||f_{\theta*} - f_0||_\infty^2)/2\sigma_\epsilon^2$$

$$\leqslant C_1'(nr_n^* + ||f_{\theta*} - f_0||_\infty^2)), \text{ w.h.p.,}$$

which concludes this lemma together with (1). $\qquad \square$

## 1.2 Proof of Lemma 4.2

Under Condition 4.1 - 4.2, we have the following lemma that shows the existence of testing functions over $\Theta_n = \Theta(L, \boldsymbol{p}, s_n)$, where $\Theta(L, \boldsymbol{p}, s_n)$ denotes the set of parameter whose $L_0$ norm is bounded by $s_n$.

**Lemma 1.2.** *Let $\varepsilon_n^* = Mn^{-1/2}\sqrt{(L+1)s^* \log N + s^* \log(p\sqrt{n/s^*})} \log^\delta(n)$ for any $\delta > 1$ and some large constant M. Let $s_n = s^* \log^{2\delta-1} n$. Then there exists some testing function $\phi \in [0, 1]$ and $C_1 > 0$, $C_2 > 1/3$, such that*

$$\mathbb{E}_{P_0}(\phi) \leqslant \exp\{-C_1 n\varepsilon_n^{*2}\},$$

$$\sup_{\substack{P_\theta \in \mathcal{F}(L, \boldsymbol{p}, s_n) \\ d(P_\theta, P_0) > \varepsilon_n^*}} \mathbb{E}_{P_\theta}(1 - \phi) \leqslant \exp\{-C_2 nd^2(P_0, P_\theta)\}.$$

*Proof.* Due to the well-known result (e.g., Le Cam (1986), page 491 or Ghosal and Van Der Vaart (2007), Lemma 2), there always exists a function $\psi \in [0, 1]$, such that

$$\mathbb{E}_{P_0}(\psi) \leqslant \exp\{-nd^2(P_{\theta_1}, P_0)/2\},$$

$$\mathbb{E}_{P_\theta}(1 - \psi) \leqslant \exp\{-nd^2(P_{\theta_1}, P_0)/2\},$$

for all $P_\theta \in \mathcal{F}(L, \boldsymbol{p}, s_n)$ satisfying that $d(P_\theta, P_{\theta_1}) \leqslant d(P_0, P_{\theta_1})/18$.

Let $K = N(\varepsilon_n^*/19, \mathcal{F}(L, \boldsymbol{p}, s_n), d(\cdot, \cdot))$ denote the covering number of set $\mathcal{F}(L, \boldsymbol{p}, s_n)$, i.e., there exists $K$ Hellinger-balls with radius $\varepsilon_n^*/19$, that completely cover $\mathcal{F}(L, \boldsymbol{p}, s_n)$. For any $\theta \in \mathcal{F}(L, \boldsymbol{p}, s_n)$

(W.O.L.G, we assume $P_\theta$ belongs to the $k$th Hellinger ball centered at $P_{\theta_k}$), if $d(P_\theta, P_0) > \varepsilon_n^*$, then we must have that $d(P_0, P_{\theta_k}) > (18/19)\varepsilon_n^*$ and there exists a testing function $\psi_k$, such that

$$
\begin{aligned}
\mathbb{E}_{P_0}(\psi_k) &\leqslant \exp\{-nd^2(P_{\theta_k}, P_0)/2\} \\
&\leqslant \exp\{-(18^2/19^2/2)n\varepsilon_n^{*2}\}, \\
\mathbb{E}_{P_\theta}(1 - \psi_k) &\leqslant \exp\{-nd^2(P_{\theta_k}, P_0)/2\} \\
&\leqslant \exp\{-n(d(P_0, P_\theta) - \varepsilon_n^*/19)^2/2\} \\
&\leqslant \exp\{-(18^2/19^2/2)nd^2(P_0, P_\theta)\}.
\end{aligned}
$$

Now we define $\phi = \max_{k=1,\ldots,K} \psi$. Thus we must have

$$
\begin{aligned}
\mathbb{E}_{P_0}(\phi) &\leqslant \sum_k \mathbb{E}_{P_0}(\psi_k) \leqslant K \exp\{-(18^2/19^2/2)n\varepsilon_n^{*2}\} \\
&\leqslant \exp\{-(18^2/19^2/2)n\varepsilon_n^{*2} - \log K\}.
\end{aligned}
$$

Note that

$$
\begin{aligned}
\log K &= \log N(\varepsilon_n^*/19, \mathcal{F}(L, \boldsymbol{p}, s_n), d(\cdot, \cdot)) \\
&\leqslant \log N(\sqrt{8}\sigma_\varepsilon \varepsilon_n^*/19, \mathcal{F}(L, \boldsymbol{p}, s_n), \|\cdot\|_\infty) \\
&\leqslant (s_n + 1)\log(\frac{38}{\sqrt{8}\sigma_\varepsilon \varepsilon_n^*}(L+1)(N+1)^{2(L+1)}) \\
&\leqslant C_0(s_n \log \frac{1}{\varepsilon_n^*} + s_n \log(L+1) + s_n(L+1)\log N) \\
&\leqslant s_n(L+1)\log n \log N \leqslant s^*(L+1)\log N \log^{2\delta} n \\
&\leqslant n\varepsilon_n^{*2}/4, \text{ for sufficiently large n}, \tag{4}
\end{aligned}
$$

where $C_0$ is some positive constant, the first inequality is due to the fact

$$
d^2(P_\theta, P_0) \leqslant 1 - \exp\{-\frac{1}{8\sigma_\epsilon^2}\|f_0 - f_\theta\|_\infty^2\}
$$

and $\varepsilon_n^* = o(1)$, the second inequality is due to Lemma 10 of Schmidt-Hieber (2017)[1], and the last inequality is due to $s_n \log(1/\varepsilon_n^*) \asymp s_n \log n$. Therefore,

$$
\mathbb{E}_{P_0}(\phi) \leqslant \sum_k P_0(\psi_k) \leqslant \exp\{-C_1 n\varepsilon_n^{*2}\},
$$

for some $C_1 = 18^2/19^2/2 - 1/4$. On the other hand, for any $\theta$, such that $d(P_\theta, P_0) \geqslant \varepsilon_n^*$, say $P_\theta$ belongs to the $k$th Hellinger ball, then we have

$$
\mathbb{E}_{P_\theta}(1 - \phi) \leqslant \mathbb{E}_{P_\theta}(1 - \psi_k) \leqslant \exp\{-C_2 nd^2(P_0, P_\theta)\},
$$

where $C_2 = 18^2/19^2/2$. Hence we conclude the proof. $\qquad \square$

Lemma 1.3 restates the Donsker and Varadhan's representation for the KL divergence, whose proof can be found in Boucheron et al. (2013).

**Lemma 1.3.** *For any probability measure $\mu$ and any measurable function $h$ with $e^h \in L_1(\mu)$,*

$$
\log \int e^{h(\eta)} \mu(d\eta) = \sup_\rho \left[ \int h(\eta) \rho(d\eta) - KL(\rho\|\mu) \right].
$$

**Proof of Lemma 4.2**

*Proof.* Denote $\Theta_n$ as the truncated parameter space $\{\theta : \sum_{i=1}^T 1(\theta_i \neq 0) \leqslant s_n\}$, where $s_n$ is defined in Lemma 1.2. Noting that

$$\int_{\theta \in \Theta} d^2(P_\theta, P_0)\widehat{q}(\theta)d\theta = \int_{\theta \in \Theta_n} d^2(P_\theta, P_0)\widehat{q}(\theta)d\theta + \int_{\theta \in \Theta_n^c} d^2(P_\theta, P_0)\widehat{q}(\theta)d\theta, \qquad (5)$$

it suffices to find upper bounds of the two components in RHS of (5).

We start with the first component. Denote $\widetilde{\pi}(\theta)$ to be the truncated prior $\pi(\theta)$ on set $\Theta_n$, i.e., $\widetilde{\pi}(\theta) = \pi(\theta)1(\theta \in \Theta_n)/\pi(\Theta_n)$, then by Lemma 1.2 and the same argument used in Theorem 3.1 of Pati et al. (2018), it could be shown

$$\int_{\Theta_n} \eta(P_\theta, P_0)\widetilde{\pi}(\theta)d\theta \leqslant e^{C_0 n \varepsilon_n^{*2}}, \text{w.h.p.} \qquad (6)$$

for some $C_0 > 0$, where $\log \eta(P_\theta, P_0) = l_n(P_\theta, P_0) + \frac{n}{3}d^2(P_\theta, P_0)$. We further denote the $\widehat{q}(\theta)$ restricted on $\Theta_n$ as $\widecheck{q}(\theta)$, i.e., $\widecheck{q}(\theta) = \widehat{q}(\theta)1(\theta \in \Theta_n)/\widehat{q}(\Theta_n)$, then by Lemma 1.3 and (6), w.h.p.,

$$\frac{n}{3\widehat{q}(\Theta_n)} \int_{\Theta_n} d^2(P_\theta, P_0)\widehat{q}(\theta)d\theta = \frac{n}{3} \int_{\Theta_n} d^2(P_\theta, P_0)\widecheck{q}(\theta)d\theta$$

$$\leqslant Cn\varepsilon_n^{*2} + \text{KL}(\widecheck{q}(\theta)||\widetilde{\pi}(\theta)) - \int_{\Theta_n} l_n(P_\theta, P_0)\widecheck{q}(\theta)d\theta. \qquad (7)$$

Furthermore,

$$\text{KL}(\widecheck{q}(\theta)||\widetilde{\pi}(\theta)) = \frac{1}{\widehat{q}(\Theta_n)} \int_{\theta \in \Theta_n} \log \frac{\widehat{q}(\theta)}{\pi(\theta)}\widehat{q}(\theta)d\theta + \log \frac{\pi(\Theta_n)}{\widehat{q}(\Theta_n)}$$

$$= \frac{1}{\widehat{q}(\Theta_n)}\text{KL}(\widehat{q}(\theta)||\pi(\theta)) - \frac{1}{\widehat{q}(\Theta_n)} \int_{\theta \in \Theta_n^c} \log \frac{\widehat{q}(\theta)}{\pi(\theta)}\widehat{q}(\theta)d\theta + \log \frac{\pi(\Theta_n)}{\widehat{q}(\Theta_n)},$$

and similarly,

$$\int_{\Theta_n} l_n(P_\theta, P_0)\widecheck{q}(\theta)d\theta = \frac{1}{\widehat{q}(\Theta_n)} \int_{\Theta} l_n(P_\theta, P_0)\widehat{q}(\theta)d\theta - \frac{1}{\widehat{q}(\Theta_n)} \int_{\Theta_n^c} l_n(P_\theta, P_0)\widehat{q}(\theta)d\theta.$$

Combining the above two equations together, we have

$$\frac{n}{3\widehat{q}(\Theta_n)} \int_{\Theta_n} d^2(P_\theta, P_0)\widehat{q}(\theta)d\theta \leqslant Cn\varepsilon_n^{*2} + \text{KL}(\widecheck{q}(\theta)||\widetilde{\pi}(\theta)) - \int_{\Theta_n} l_n(P_\theta, P_0)\widecheck{q}(\theta)d\theta$$

$$= Cn\varepsilon_n^{*2} + \frac{1}{\widehat{q}(\Theta_n)}\left(\text{KL}(\widehat{q}(\theta)||\pi(\theta)) - \int_{\Theta} l_n(P_\theta, P_0)\widehat{q}(\theta)d\theta\right) \qquad (8)$$

$$- \frac{1}{\widehat{q}(\Theta_n)}\left(\int_{\Theta_n^c} \log \frac{\widehat{q}(\theta)}{\pi(\theta)}\widehat{q}(\theta)d\theta - \int_{\Theta_n^c} l_n(P_\theta, P_0)\widehat{q}(\theta)d\theta\right) + \log \frac{\pi(\Theta_n)}{\widehat{q}(\Theta_n)}.$$

The second component of (5) trivially satisfies that $\int_{\theta \in \Theta_n^c} d^2(P_\theta, P_0)\widehat{q}(\theta)d\theta \leqslant \int_{\theta \in \Theta_n^c} \widehat{q}(\theta)d\theta = \widehat{q}(\Theta_n^c)$. Thus, together with (8), we have that w.h.p.,

$$\int d^2(P_\theta, P_0)\widehat{q}(\theta)d\theta \leqslant 3\widehat{q}(\Theta_n)C\varepsilon_n^{*2} + \frac{3}{n}\left(\text{KL}(\widehat{q}(\theta)||\pi(\theta)) - \int_{\Theta} l_n(P_\theta, P_0)\widehat{q}(\theta)d\theta\right)$$

$$+ \frac{3}{n} \int_{\Theta_n^c} l_n(P_\theta, P_0)\widehat{q}(\theta)d\theta + \frac{3}{n} \int_{\Theta_n^c} \log \frac{\pi(\theta)}{\widehat{q}(\theta)}\widehat{q}(\theta)d\theta + \frac{3\widehat{q}(\Theta_n)}{n} \log \frac{\pi(\Theta_n)}{\widehat{q}(\Theta_n)} + \widehat{q}(\Theta_n^c). \qquad (9)$$

The second term in the RHS of (9) is bounded by $C'(r_n^* + \xi_n^*)$ w.h.p., due to Lemma 4.1, where $C'$ is either positive constant or diverging sequence depending on whether $n(r_n^* + \xi_n^*)$ diverges.

The third term in the RHS of (9) is bounded by

$$\frac{3}{n}\int_{\Theta_n^c} l_n(P_\theta, P_0)\widehat{q}(\theta)d\theta$$

$$=\frac{3}{2n\sigma_\epsilon^2}\int_{\Theta_n^c}\left[\sum_{i=1}^n \epsilon_i^2 - \sum_{i=1}^n (\epsilon_i + f_0(X_i) - f_\theta(X_i))^2\right]\widehat{q}(\theta)d\theta$$

$$=\frac{3}{2n\sigma_\epsilon^2}\int_{\Theta_n^c}\left[-2\sum_{i=1}^n(\epsilon_i \times (f_0(X_i) - f_\theta(X_i)) - \sum_{i=1}^n(f_0(X_i) - f_\theta(X_i))^2\right]\widehat{q}(\theta)d\theta$$

$$=\frac{3}{2n\sigma_\epsilon^2}\left\{-2\sum_{i=1}^n\epsilon_i\int_{\Theta_n^c}(f_0(X_i) - f_\theta(X_i))\widehat{q}(\theta)d\theta - \int_{\Theta_n^c}\sum_{i=1}^n(f_0(X_i) - f_\theta(X_i))^2\widehat{q}(\theta)d\theta\right\}.$$

Conditional on $X_i$'s, $-2\sum_{i=1}^n \epsilon_i \int_{\Theta_n^c}(f_0(X_i) - f_\theta(X_i))\widehat{q}(\theta)d\theta$ follows a normal distribution $\mathcal{N}(0, V^2)$, where $V^2 = 4\sigma_\epsilon^2\sum_{i=1}^n(\int_{\Theta_n^c}(f_0(X_i) - f_\theta(X_i))\widehat{q}(\theta)d\theta)^2 \leqslant 4\sigma_\epsilon^2\int_{\Theta_n^c}\sum_{i=1}^n(f_0(X_i) - f_\theta(X_i))^2\widehat{q}(\theta)d\theta$. Thus conditional on $X_i$'s, the third term in the RHS of (9) is bounded by

$$\frac{3}{2n\sigma_\epsilon^2}\left[\mathcal{N}(0, V^2) - \frac{V^2}{4\sigma_\epsilon^2}\right]. \tag{10}$$

Noting that $\mathcal{N}(0, V^2) = O_p(M_n V)$ for any diverging sequence $M_n$, (10) is further bounded, w.h.p., by

$$\frac{3}{2n\sigma_\epsilon^2}(M_n V - \frac{V^2}{4\sigma_\epsilon^2}) \leqslant \frac{3}{2n\sigma_\epsilon^2}\sigma_\epsilon^2 M_n^2.$$

Therefore, the third term in the RHS of (9) can be bounded by $\varepsilon_n^{*2}$ w.h.p. (by choosing $M_n^2 = n\varepsilon_n^{*2}$).

The fourth term in the RHS of (9) is bounded by

$$\frac{3}{n}\int_{\Theta_n^c}\log\frac{\pi(\theta)}{\widehat{q}(\theta)}\widehat{q}(\theta)d\theta \leqslant \frac{3}{n}\widehat{q}(\Theta_n^c)\log\frac{\pi(\Theta_n^c)}{\widehat{q}(\Theta_n^c)} \leqslant \frac{3}{n}\sup_{x\in(0,1)}[x\log(1/x)] = O(1/n).$$

Similarly, the fifth term in the RHS of (9) is bounded by $O(1/n)$.

For the last term in the RHS of (9), by Lemma 1.5 in below, w.h.p., $\widehat{q}(\Theta_n^c) \leqslant \varepsilon_n^{*2}$.

Combine all the above result together, w.h.p.,

$$\int d^2(P_\theta, P_0)\widehat{q}(\theta)d\theta \leqslant C\varepsilon_n^{*2} + \frac{3}{n}\left(\mathrm{KL}(\widehat{q}(\theta)||\pi(\theta)) - \int_\Theta l_n(P_\theta, P_0)\widehat{q}(\theta)d\theta\right) + O(1/n),$$

where $C$ is some constant. $\qquad\square$

**Lemma 1.4** (Chernoff bound for Poisson tail). *Let $X \sim poi(\lambda)$ be a Poisson random variable. For any $x > \lambda$,*

$$P(X \geqslant x) \leqslant \frac{(e\lambda)^x e^{-\lambda}}{x^x}.$$

**Lemma 1.5.** *If $\lambda \leqslant T^{-1}\exp\{-Mnr_n^*/s_n\}$ for any positive diverging sequence $M \to \infty$, then w.h.p., $\widehat{q}(\Theta_n^c) = O(\varepsilon_n^{*2})$.*

*Proof.* By Lemma 4.1, we have that w.h.p.,

$$\mathrm{KL}(\widehat{q}(\theta)||\pi(\theta|\lambda)) + \int_\Theta l_n(P_0, P_\theta)\widehat{q}(\theta)d\theta = \inf_{q_\theta \in \mathcal{Q}}\left\{\mathrm{KL}(q(\theta)||\pi(\theta|\lambda)) + \int_\Theta l_n(P_0, P_\theta)q(\theta)(d\theta)\right\}$$

$$\leqslant Cnr_n^* \quad \text{(Note that } r_n^* \asymp \xi_n^*\text{)}$$

where $C$ is either a constant or any diverging sequence, depending on whether $nr_n^*$ diverges. By the similar argument used in the proof of Lemma 4.1,

$$\int_\Theta l_n(P_0, P_\theta)\widehat{q}(\theta)d\theta \leqslant \frac{1}{2\sigma_\epsilon^2}\left(\int_\Theta ||f_\theta(X) - f_0(X)||_2^2\widehat{q}(\theta)(d\theta) + Z\right)$$

where $Z$ is a normal distributed $\mathcal{N}(0, \sigma_\epsilon^2 c_0')$, where $c_0' \leqslant c_0 = \int_\Theta \|f_\theta(X) - f_0(X)\|_2^2 \widehat{q}(\theta)(d\theta)$. Therefore, $-\int_\Theta l_n(P_0, P_\theta)\widehat{q}(\theta)d\theta = (1/2\sigma_\epsilon^2)[-c_0 + O_p(\sqrt{c_0})]$, and $\mathrm{KL}(\widehat{q}(\theta)\|\pi(\theta|\lambda)) \leqslant Cnr_n^* + (1/2\sigma_\epsilon^2)[-c_0 + O_p(\sqrt{c_0})]$. Since $Cnr_n^* \to \infty$, we must have w.h.p., $\mathrm{KL}(\widehat{q}(\theta)\|\pi(\theta|\lambda)) \leqslant Cnr_n^*/2$. On the other hand,

$$
\begin{aligned}
\mathrm{KL}(\widehat{q}(\theta)\|\pi(\theta|\lambda)) &= \sum_{i=1}^T \mathrm{KL}(\widehat{q}(\theta_i)\|\pi(\theta_i|\lambda)) \geqslant \sum_{i=1}^T \mathrm{KL}(\widehat{q}(\gamma_i)\|\pi(\gamma_i|\lambda)) \\
&= \sum_{i=1}^T \left[ \widehat{q}(\gamma_i = 1) \log \frac{\widehat{q}(\gamma_i = 1)}{\lambda} + \widehat{q}(\gamma_i = 0) \log \frac{\widehat{q}(\gamma_i = 0)}{1 - \lambda} \right].
\end{aligned}
\tag{11}
$$

Let us choose $\lambda_0 = 1/T$, and $A = \{i : \widehat{q}(\gamma_i = 1) \geqslant \lambda_0\}$, then the above inequality (11) implies that $\sum_{i \in A} \widehat{q}(\gamma_i = 1) \log(\lambda_0/\lambda) \leqslant Cnr_n^*/2$. Noting that $\lambda \leqslant T^{-1} \exp\{-Mnr_n^*/s_n\}$, it further implies $\sum_{i \in A} \widehat{q}(\gamma_i = 1) \leqslant s_n/M \prec s_n$.

Under distribution $\widehat{q}$, by Bernstein inequality,

$$
\begin{aligned}
Pr(\sum_{i \in A} \gamma_i \geqslant 2s_n/3) &\leqslant Pr(\sum_{i \in A} \gamma_i \geqslant s_n/2 + \sum_{i \in A} \mathbb{E}(\gamma_i)) \leqslant \exp\left( -\frac{s_n^2/8}{\sum_{i \in A} \mathbb{E}[\gamma_i^2] + s_n/6} \right) \\
&= \exp\left( -\frac{s_n^2/8}{\sum_{i \in A} \widehat{q}(\gamma_i = 1) + s_n/6} \right) \leqslant \exp\left( -cs_n \right) = O(\varepsilon_n^{*2})
\end{aligned}
$$

for some constant $c > 0$, where the last inequality holds since $\log(1/\varepsilon_n^{*2}) = O(\log n) \prec s_n$.

Under distribution $\widehat{q}$, $\sum_{i \notin A} \gamma_i$ is stochastically smaller than $Bin(T, \lambda_0)$. Since $T \to \infty$, then by Lemma 1.4,

$$
Pr(\sum_{i \notin A} \gamma_i \geqslant s_n/3) \leqslant Pr(Bin(T, \lambda_0) \geqslant s_n/3) \to Pr(\mathrm{poi}(1) \geqslant s_n/3)
$$

$$
= O(\exp\{-c's_n\}) = O(\varepsilon_n^{*2})
$$

for some $c' > 0$. Trivially, it implies that w.h.p, $Pr(\sum_i \gamma_i \geqslant s_n) = O(\varepsilon_n^{*2})$ for VB posterior $\widehat{q}$. $\qquad \square$

## 1.3 Main theorem

**Theorem 1.1.** *Under Conditions 4.1-4.2, 4.4 and set $-\log \lambda = \log(T) + \delta[(L+1)\log N + \log \sqrt{np}]$ for any constant $\delta > 0$, we then have that w.h.p.,*

$$
\int_\Theta d^2(P_\theta, P_0)\widehat{q}(\theta)d\theta \leqslant C\varepsilon_n^{*2} + C'(r_n^* + \xi_n^*),
$$

*where $C$ is some positive constant and $C'$ is any diverging sequence. If $\|f_0\|_\infty < F$, and we truncated the VB posterior on $\Theta_F = \{\theta : \|f_\theta\|_\infty \leqslant F\}$, i.e., $\widehat{q}_F \propto \widehat{q}1(\theta \in \Theta_F)$, then, w.h.p.,*

$$
\int_{\Theta_F} \mathbb{E}_X|f_\theta(X) - f_0(X)|^2 \widehat{q}_F(\theta)d\theta \leqslant \frac{C\varepsilon_n^{*2} + C'(r_n^* + \xi_n^*)}{C_F \widehat{q}(\Theta_F)}
$$

*where $C_F = [1 - \exp(-4F^2/8\sigma_\epsilon^2)]/4F^2$, and $\widehat{q}(\Theta_F)$ is the VB posterior mass of $\Theta_F$.*

*Proof.* The convergence under squared Hellinger distance is directly result of Lemma 4.1 and 4.2, by simply checking the choice of $\lambda$ satisfies required conditions. The convergence under $L_2$ distance relies on inequality $d^2(P_\theta, P_0) \geqslant C_F \mathbb{E}_X|f_\theta(X) - f_0(X)|^2$ for $C_F = [1 - \exp(-4F^2/8\sigma_\epsilon^2)]/4F^2$ when both $f_\theta$ and $f_0$ are bounded by $F$. Then, w.h.p,

$$
\begin{aligned}
\int_{\Theta_F} \mathbb{E}_X|f_\theta(X) - f_0(X)|^2 \widehat{q}_F(\theta)d\theta &\leqslant C_F^{-1} \int_{\Theta_F} d^2(P_\theta, P_0)\widehat{q}_F(\theta)d\theta \\
&\leqslant \frac{1}{C_F \widehat{q}(\Theta_F)} \int_\Theta d^2(P_\theta, P_0)\widehat{q}(\theta)d\theta \leqslant \frac{C\varepsilon_n^{*2} + C'(r_n^* + \xi_n^*)}{C_F \widehat{q}(\Theta_F)}.
\end{aligned}
$$

$\qquad \square$

## 2 Additional experimental results

### 2.1 Comparison between Bernoulli variable and the Gumbel softmax approximation

Denote $\gamma_i \sim \text{Bern}(\phi_i)$ and $\widetilde{\gamma}_i \sim \text{Gumbel-softmax}(\phi_i, \tau)$, then we have that

$$\widetilde{\gamma}_i := g_\tau(\phi_i, u_i) = (1 + \exp(-\eta_i/\tau))^{-1}, \quad \text{where } \eta_i = \log \frac{\phi_i}{1 - \phi_i} + \log \frac{u_i}{1 - u_i}, \quad u_i \sim \mathcal{U}(0, 1),$$

$$\gamma_i := g(\phi_i, u_i) = 1(u_i \leqslant \phi_i) \quad \text{where } u_i \sim \mathcal{U}(0, 1).$$

Fig 1 demonstrates the functional convergence of $g_\tau$ towards $g$ as $\tau$ goes to zero. In Fig 1(a), by fixing $\phi_i(= 0.9)$, we show $g_\tau$ converges to $g$ as a function of $u_i$. Fig 1 (b) demonstrates that $g_\tau$ converges to $g$ as a function of $\alpha_i = \log(\phi_i/(1 - \phi_i))$ when $u_i(= 0.2)$ is fixed. These two figures show that as $\tau \to 0$, $g_\tau \to g$. Formally, Maddison et al. (2017) rigorously proved that $\widetilde{\gamma}_i$ converges to $\gamma_i$ in distribution as $\tau$ approaches 0.

(a) Fix $\phi_i = 0.9$.                    (b) Fix $u_i = 0.2$.

Figure 1: The convergence of $g_\tau$ towards $g$ as $\tau$ approaches 0.

### 2.2 Algorithm implementation details for the numerical experiments

**Initialization**    As mentioned by Sønderby et al. (2016) and Molchanov et al. (2017), training sparse BNN with random initialization may lead to bad performance, since many of the weights could be pruned too early. In our case, we assign each of the weights and biases a inclusion variable, which could reduce to zero quickly in the early optimization stage if we randomly initialize them. As a consequence, we deliberately initialize $\phi_i$ to be close to 1 in our experiments. This initialization strategy ensures the training starts from a fully connected neural network, which is similar to start training from a pre-trained fully connected network as mentioned in Molchanov et al. (2017). The other two parameters $\mu_i$ and $\sigma_i$ are initialized randomly.

**Other implementation details in simulation studies**    We set $K = 1$ and learning rate $= 5 \times 10^{-3}$ during training. For Simulation I, we choose batch size $m = 1024$ and $m = 128$ for (A) and (B) respectively, and run 10000 epochs for both cases. For simulation II, we use $m = 512$ and run 7000 epochs. Although it is common to set up an annealing schedule for temperature parameter $\tau$, we don't observe any significant performance improvement compared to setting $\tau$ as a constant, therefore we choose $\tau = 0.5$ in all of our experiments. The optimization method used is Adam.

The implementation details for UCI datasets and MNIST can be found in Section 2.5 and 2.6 respectively.

### 2.3 Toy example: linear regression

In this section, we aim to demonstrate that there is little difference between the results using inverse-CDF reparameterization and Gumbel-softmax approximation via a toy example.

Consider a linear regression model:

$$Y_i = X_i^T \beta + \epsilon_i, \quad \epsilon_i \sim \mathcal{N}(0, 1), \quad i = 1, \ldots, n,$$

We simulate a dataset with 1000 observations and 200 predictors, where $\beta_{50} = \beta_{100} = \beta_{150} = 10$, $\beta_{75} = \beta_{125} = -10$ and $\beta_j = 0$ for all other $j$.

A spike-and-slab prior is imposed on $\beta$ such that

$$\beta_j|\gamma_j \sim \gamma_j \mathcal{N}(0, \sigma_0^2) + (1 - \gamma_j)\delta_0, \quad \gamma_j \sim \text{Bern}(\lambda),$$

for $j = 1, \ldots, 200$, where $\sigma_0 = 5$ and $\lambda = 0.03$. The variational distribution $q(\beta)\mathcal{Q}$ is chosen as

$$\beta_j|\gamma_j \sim \gamma_j \mathcal{N}(\mu_j, \sigma_j^2) + (1 - \gamma_j)\delta_0, \quad \gamma_j \sim \text{Bern}(\phi_j).$$

We use both Gumbel-softmax approximation and inverse-CDF reparameterization for the stochastic optimization of ELBO, and plot posterior mean $\mathbb{E}_{\hat{q}(\beta)}(\beta_j|\gamma_j)$ (blue curve) against the true value (red curve). Figure 2 shows that inverse-CDF reparameterization exhibits only slightly higher error in estimating zero coefficients than the Gumbel-softmax approximation, which indicates the two methods has little difference on this toy example.

(a) Gumbel-softmax reparametrization           (b) Inverse-CDF reparametrization

Figure 2: Linear regression

## 2.4 Teacher student networks

The network parameter $\theta$ for the sparse teacher network setting (B) is set as following: $W = \{W_{1,11} = W_{1,12} = W_{2,11} = W_{2,12} = 2.5, W_{1,21} = W_{1,22} = W_{2,21} = W_{2,22} = 1.5, W_{3,11} = 3$ and $W_{3,21} = 2\}$; $b = \{b_{1,1} = b_{2,1} = b_{3,1} = 1$ and $b_{1,2} = b_{2,2} = -1\}$.

Figure 3 displays the simulation result for simulation I under dense teacher network (A) setting. Unlike the result under sparse teacher network (B), the testing accuracy seems monotonically increases as $\lambda$ increases (i.e., posterior network gets denser). However, as shown, the increasing of testing performance is rather slow, which indicates that introducing sparsity has few negative impact to the testing accuracy.

**Coverage rate**   In this paragraph, we explain the details of how we compute the coverage rate values of Bayesian intervals reported in the main text. A fixed point $(x_1^{(*)}, \ldots, x_p^{(*)})'$ is prespecified, and let $x^{(1)}, \ldots, x^{(1000)}$ be 1000 equidistant points from $-1$ to $1$. In each run, we compute the Bayesian credible intervals of response means (estimated by 600 Monte Carlo samples) for 1000 different input $x$'s: $(x^{(1)}, x_2^{(*)}, \ldots, x_p^{(*)}), \ldots, (x^{(1000)}, x_2^{(*)}, \ldots, x_p^{(*)})$. It is repeated by 60 times and the average coverage rate (over all different $x$'s and 60 runs) is reported. Similarly, we replace $x_2^{(*)}$ (or $x_3^{(*)}$) by $x^{(i)}$ ($i = 1, \ldots, 1000$), and compute their average coverage rate. The complete coverage rate results are shown in Table 1. Note that Table 1 in the main text shows $95\%$ coverage of $x_3$ for (A) and $95\%$ coverage of $x_1$ for (B).

## 2.5 Real data regression experiment: UCI datasets

We follow the experimental protocols of Hernández-Lobato and Adams (2015), and choose five datasets for the experiment. For the small datasets "Kin8nm", "Naval", "Power Plant" and "wine", we choose a single-hidden-layer ReLU network with 50 hidden units. We randomly select $90\%$ and $10\%$ for training and testing respectively, and this random split process is repeated for 20 times (to obtain standard deviations for our results). We choose minibatch size $m = 128$, learning rate $= 10^{-3}$ and

(a) $\lambda \leqslant \lambda_{opt}$.      (b) $\lambda \geqslant \lambda_{opt}$.      (c) Dense teacher network.

Figure 3: (a) $\lambda = \{10^{-200}, 10^{-150}, 10^{-100}, 10^{-50}, 10^{-20}, 10^{-5}, \lambda_{opt}\}$. (b) $\lambda = \{\lambda_{opt}, 0.1, 0.3, 0.5, 0.7, 0.9, 0.99\}$. (c) The structure of the target dense teacher network.

Table 1: Coverage rates for teacher networks.

| | | 90 % coverage (%) | | | 95% coverage (%) | | |
|---|---|---|---|---|---|---|---|
| | **Method** | $x_1$ | $x_2$ | $x_3$ | $x_1$ | $x_2$ | $x_3$ |
| Dense | SVBNN | $93.8 \pm 2.84$ | $93.1 \pm 4.93$ | $93.1 \pm 2.96$ | $97.9 \pm 1.01$ | $97.9 \pm 1.69$ | $97.5 \pm 1.71$ |
| | VBNN | $85.8 \pm 2.51$ | $82.4 \pm 2.62$ | $86.3 \pm 1.88$ | $92.7 \pm 2.83$ | $91.3 \pm 2.61$ | $91.4 \pm 2.43$ |
| | VD | $61.3 \pm 2.40$ | $60.0 \pm 2.79$ | $64.9 \pm 6.17$ | $74.9 \pm 1.79$ | $71.8 \pm 2.33$ | $76.4 \pm 4.75$ |
| | HS-BNN | $83.1 \pm 1.67$ | $80.0 \pm 1.21$ | $76.9 \pm 1.70$ | $88.1 \pm 1.13$ | $84.1 \pm 1.48$ | $83.5 \pm 0.78$ |
| Sparse | SVBNN | $92.3 \pm 8.61$ | $94.6 \pm 5.37$ | $98.3 \pm 0.00$ | $96.4 \pm 4.73$ | $97.7 \pm 3.71$ | $100 \pm 0.00$ |
| | VBNN | $86.7 \pm 10.9$ | $87.0 \pm 11.3$ | $93.3 \pm 0.00$ | $90.7 \pm 8.15$ | $91.9 \pm 9.21$ | $96.7 \pm 0.00$ |
| | VD | $65.2 \pm 0.08$ | $63.7 \pm 6.58$ | $65.9 \pm 0.83$ | $75.5 \pm 7.81$ | $74.6 \pm 7.79$ | $76.6 \pm 0.40$ |
| | HS-BNN | $59.0 \pm 8.52$ | $59.4 \pm 4.38$ | $56.6 \pm 2.06$ | $67.0 \pm 8.54$ | $68.2 \pm 3.62$ | $66.5 \pm 1.86$ |

run 500 epochs for "Naval", "Power Plant" and "Wine", 800 epochs for "Kin8nm". For the large dataset "Year", we use a single-hidden-layer ReLU network with 100 hidden units, and the evaluation is conducted on a single split. We choose $m = 256$, learning rate $= 10^{-3}$ and run 100 epochs. For all the five datasets, $\lambda$ is chosen as $\lambda_{opt}$: $\log(\lambda_{opt}^{-1}) = \log(T) + 0.1[(L+1)\log N + \log\sqrt{n}p]$, which is the same as other numerical studies. We let $\sigma_0^2 = 2$ and use grid search to find $\sigma_\epsilon$ that yields the best prediction accuracy. Adam is used for all the datasets in the experiment.

We report the testing squared root MSE (RMSE) based on $\widehat{f}_H$ (defined in the main text) with $H = 30$, and also report the posterior network sparsity $\widehat{s} = \sum_{i=1}^{T} \phi_i / T$. For the purpose of comparison, we list the results by Horseshoe BNN (HS-BNN) (Ghosh and Doshi-Velez, 2017) and probalistic backpropagation (PBP) (Hernández-Lobato and Adams, 2015). Table 2 demonstrates that our method achieves best prediction accuracy for all the datasets with a sparse structure.

Table 2: Results on UCI regression datasets.

| Dataset | $n(p)$ | Test RMSE | | | Posterior sparsity(%) |
|---|---|---|---|---|---|
| | | SVBNN | HS-BNN | PBP | SVBNN |
| Kin8nm | 8192 (8) | $0.08 \pm 0.00$ | $0.08 \pm 0.00$ | $0.10 \pm 0.00$ | $64.5 \pm 1.85$ |
| Naval | 11934 (16) | $0.00 \pm 0.00$ | $0.00 \pm 0.00$ | $0.01 \pm 0.00$ | $82.9 \pm 1.31$ |
| Power Plant | 9568 (4) | $4.01 \pm 0.18$ | $4.03 \pm 0.15$ | $4.12 \pm 0.03$ | $56.6 \pm 3.13$ |
| Wine | 1599 (11) | $0.62 \pm 0.04$ | $0.63 \pm 0.04$ | $0.64 \pm 0.01$ | $59.9 \pm 4.92$ |
| Year | 515345 (90) | $8.87 \pm$ NA | $9.26 \pm$ NA | $8.88 \pm$ NA | $20.8 \pm$ NA |

## 2.6 Real data classification experiment: MNIST dataset

The MNIST data is normalized by mean equaling 0.1306 and standard deviation equaling 0.3081. For all methods, we choose the same minibatch size $m = 256$, learning rate $= 5 \times 10^{-3}$ for our method and $3 \times 10^{-3}$ for the others, total number of epochs is 400 and the optimization algorithm is RMSprop. AGP is pre-specified at $5\%$ sparsity level.

| (a) Overlaid images (on the last column) | (b) Predictive distribution for overlaid images |

Figure 4: Top row of (b) exhibits the predictive distribution for the top overlaid image, which is made by 5 and 6; Middle row of (b) exhibits the predictive distribution for the middle overlaid image, which is made by 2 and 3; Bottom row of (b) exhibits the predictive distribution for the bottom overlaid image, which is made by 2 and 7.

Besides the testing accuracy reported in the main text, we also examine our method's ability of uncertainty quantification for MNIST classification task. We first create ambiguous images by overlaying two examples from the testing set as shown in Figure 4 (a). To perform uncertainty quantification using our method, for each of the overlaid images, we generate $\theta_h$ from the VB posterior $\widehat{q}(\theta)$ for $h = 1, \ldots, 100$, and calculate the associated predictive probability vector $f_{\theta_h}(x) \in \mathbb{R}^{10}$ where $x$ is the overlaid image input, and then use the estimated posterior mean $\widehat{f}(x) = \sum_{h=1}^{100} f_{\theta_h}(x)/100$ as the Bayesian predictive probability vector. As a comparison, we also calculate the predictive probability vector for each overlaid image using AGP as a frequentist benchmark. Figure 4 (b) shows frequentist method gives almost a deterministic answer (i.e., predictive probability is almost 1 for certain digit) that is obviously unsatisfactory for this task, while our VB method is capable of providing knowledge of certainty on these out-of-domain inputs, which demonstrates the advantage of Bayesian method in uncertainty quantification on the classification task.

## 2.7 Illustration of CNN: Fashion-MNIST dataset

In this section, we perform an experiment on a more complex task, the Fashion-MNIST dataset. To illustrate the usage of our method beyond feedforward networks, we consider using a 2-Conv-2-FC network: The feature maps for the convolutional layers are set to be 32 and 64, and the filter size are $5 \times 5$ and $3 \times 3$ respectively. The paddings are 2 for both layers and the it has a $2 \times 2$ max pooling for each of the layers; The fully-connected layers have $64 \times 8 \times 8$ neurons. The activation functions are all ReLUs. The dataset is prepocessed by random horizontal flip. The batchsize is 1024, learning rate is 0.001, and Adam is used for optimization. We run the experiment for 150 epochs.

We use both SVBNN and VBNN for this task. In particular, the VBNN, which uses normal prior and variational distributions, is the full Bayesian method without compressing, and can be regarded as the baseline for our method. Figure 5 exhibits our method attains higher accuracy as epoch increases and then decreases as the sparsity goes down. Meanwhile, the baseline method - full BNN suffers from overfitting after 80 epochs.

| (a) Accuracy. | (b) Sparsity. |

Figure 5: Fashion-MNIST experiment.

## Footnotes

[1]Although Schmidt-Hieber (2017) only focuses on ReLU network, its Lemma 10 could apply to any 1-Lipchitz continuous activation function.