[Reviews · NeurIPS 2020]

Review 1

Summary and Contributions: This paper introduces a spike-and-slab prior to Bayesian neural networks using variational inference. The primary focus of the work is the considerable amount of theoretical work to show the consistency of variational inference in this framework. Additional experimental work shows the effectiveness, especially for recovering true variables in a sparse regression setting with low false positive rate and low false negative rate. Update: author feedback is appreciated, and based on reading other reviews, there were some minor issues I seem to have glossed over, and am slightly lowering my score (I'm not sure the author feedback was wholly convincing, but the points are sufficiently minor), but still vote for a clear accept.

Strengths: This work incorporates a natural way of modeling sparsity into Bayesian neural networks, which has the potential to be extremely valuable – allowing for a natural means of expressing model uncertainty while simultaneously providing a natural means of model compression. The theoretical grounding is considerable, and I am unaware of other works that incorporate the spike-and-slab prior/variational posterior family in BNNs, so it is a novel work as well.

Weaknesses: I think the only real limitation of this work is that additional outcomes for success and additional experiments could have been provided, but I think the simulation and real data examples are certainly adequate, especially for a work that has a primary focus on the theoretical nature of the problem.

Correctness: As far as I am aware there are no problems with the method or claims.

Clarity: The paper is indeed well-written, I noticed no issues. The notation is occasionally a bit difficult to follow, but that's more due to the complicated nature of the problem, so I would not recommend any changes.

Relation to Prior Work: I believe prior work was well-addressed as well and clearly shows the contributions made in this work.

Reproducibility: Yes

Additional Feedback: I see no obvious areas that need improvement. Maybe some of the equations (say, equations 6 and 7) could be compressed just to save space, but this is minor at best.


Review 2

Summary and Contributions: The paper proposes to use a spike-and-slap prior to learn a sparse Bayesian Neural Network (BNN). The proposed variational posterior (again spike-and-slap) is then shown theoretically to be consistent together with a convergence rate result.

Strengths: The approach is well motivated with a theoretically well-founded evaluation.

Weaknesses: What is somewhat lacking is the empirical evaluation and a proper discussion with respect to existing approaches, how the proposed method fits in among them, and why it is needed. (See below for more detailed comments.)

Correctness: The claims and assumptions are well formulated with extensive proofs in the supplementary material and seem to be correct. The same holds for the empirical methodology.

Clarity: The paper is well written.

Relation to Prior Work: While the method itself is well defended, its relation to prior work is somewhat lacking. E.g. while Blundell et al. and Deng et al. are cited, their respective spike-and-slap variants (where the Dirac is replaced with a Gaussian with small variance in the case of Blundell and a Laplace in the case of Deng) are not really discussed and especially not evaluated against.

Reproducibility: Yes

Additional Feedback: #### Post Rebuttal Update I thank the authors for their feedback. While the two weaknesses (a proper discussion of similar approaches in the BNN literature, demonstration of the approach on more complex models) of the paper that I see still somewhat remain, I still recommend acceptance and change my score from 6 to 7. I encourage the authors to extend the theoretical discussion to related prior work in the final version as they already indicated in the rebuttal, and to pursue the question of how to extend the method to more complex models in future studies. ####################### ## Major questions/comments - As mentioned above a proper empirical evaluation and comparison with similar sparsity inducing approaches seems necessary to demonstrate that the strength of the model lies not only in the theoretical contribution but that it is also a valuable addition in practice. As mentioned above especially other spike-and-slap approaches such as Blundell et al. (sofar Blundell is only compared against in the plain Gaussian prior setting), and Deng et al.. But also other approaches such as horseshoe priors (e.g. (Louizos et al. 2017), (Ghosh et al, 2018) or the already cited (Ghosh et al.)). - Similarly, the experimental setting of the paper is constrained to rather small networks, where the sparsity motivation of the introduction is not really necessary. They serve as proper proof of concept that the approach works in principle, but do the authors have an intuition as to how well their approach scales to larger models? ## Minor questions/comments - Sparsity in itself won't help if it is not structured (e.g. dropping a whole convolutional filter helps in saving operations, while only dropping some weights in a filter while keeping the rest still keeps most operations in practice). Can the authors comment on how they assume their approach could be generalized? (E.g. in the direction of structured approaches such as (Neklyudov et al., 2017,...) - l112/113 claim Molchanov et al. to be a frequentist pruning approach. Can the authors give more details about why they do not consider the pruning approach of that paper to be Bayesian? - The regression results in the appendix are only on a subset of the commonly used UCI setup from Hernandez-Lobato and Adams. Is there a reason why those five are chosen over the others? - The authors provide all the necessary details for a reproduction of the results in the main paper & the supplementary together with code. A very minor improvement would be to mention in the respective places (Figure 1/Table 1/2) what the error bars are for completeness. (They are probably all standard deviations in the main paper? But e.g. in Table 2 in the Supplementary they are standard error instead.) ______ Ghosh et al., Structured Variational Learning of Bayesian Neural Networks with Horseshoe Priors, ICML 2018 Louizos et al., Bayesian Compression for Deep Learning, NeurIPS 2017 Neklyudov et al., Structured Bayesian Pruning via Log-Normal Multiplicative Noise, NeurIPS 2017


Review 3

Summary and Contributions: The authors proposed a sparse Bayesian deep neural network model with spike-and-slab prior. This work is motivated to close the gap between theoretical studies and practical studies in deep Bayesian neural networks. From this perspective, they provided the theoretical guarantee for the variational consistency. The authors also showed that the corresponding convergence rate strikes the balance of statistical estimation error, variational error, and the approximation error when we chose the hyperparameter of Bernoulli distribution in spike-and-slab prior. Furthermore, through several experiments, they confirmed the validity of their theoretical results and demonstrated that the proposed method achieved the better performance of variable selection and uncertainty quantification than previous work.

Strengths: ・Although the idea itself is a simple (using sparse prior; spike-and-slab prior), the author provided a valid theoretical guarantee for consistency and showed the optimal hyperparameter value of Bernoulli distribution in spike-and-slab prior, that is helpful for users and guarantees the performances of the proposed method. ・There are enough experiments to confirm the theoretical results and predictive performance, uncertainty quantification, and variable selection for multi-layer networks. ・The paper is well-written and easy to understand what they want to do and what the contribution is.

Weaknesses: Limitation : (a) : Theoretical analysis only holds when we use the Gaussian variational distribution with the same family of the spike-and-slab prior. Therefore, the performance is not guaranteed when we set the other variational distributions. (b) : They conducted the experiments on a simple dataset, e.g., MNIST. Therefore, we cannot understand whether the performance holds when we want to apply the more complex BNN model for more complicated tasks. Furthermore, the performance is not guaranteed when using the more sophisticated variational inference framework for complex BNN models, e.g., black-box VI (Ranganath, 2014) or hierarchical VI (Ranganath, 2016). On more complex models and datasets (at least, Fashion-MNIST), how much better does the proposed method perform in terms of prediction accuracy or/and uncertainty quantification compared to comparative methods? (c) : There are no experimental results for convergence speed in terms of real-world time, which is one of the open problems in BNN. Certainly, the proposed method is effective for the memory storage and computational burden, but it is not clear how fast the inference is. How fast can the proposed method conduct inference with sparsification and variable selection?

Correctness: It seems that the claims from theoretical analysis and the empirical methodology are correct and reasonable.

Clarity: The structure and organization of the presentation are good. The explanation of contribution is clear and easy to follow the related work.

Relation to Prior Work: The related work section is well-written. This paper's contribution differs from the previous contributions in terms of the variable selection (for multi-layer networks) and providing the essential theoretical analysis for consistency in Bayesian deep learning.

Reproducibility: Yes

Additional Feedback: All of the feedbacks, comments, and suggestions are in the above sections. ========================= After reading the author response ========================= I would thank the author(s) for their feedback. I read it and the other reviewers’ comments. After reading this, my concerns almost has been addressed. I still think that it is important to compare the empirical results based on real-world time, even if the authors do not have the cutting-edge infrastructure; however, it does not largely reduce the contributions of this paper. Therefore, I decided not to change my score as 7. I recommend the authors to show the more experimental results on more complicated dataset in the future version of this paper, at least, on Fashion-MNIST, as the authors reported in their feedback. Furthermore, I encourage to extend the theoretical discussion to related work, as the authors claimed in their feedback. I’m looking forward to seeing their future work for the computational efficient algorithm of the proposed approach.


Review 4

Summary and Contributions: The work investigates sparsity inducing spike-and-slap priors for training multi-layer neural networks. It derives theoretical requirements on hyperparameters of the prior distribution for posterior contraction. Finally, the work empirically shows that the devised approach leads to better recovery of ground truth sparsity in a teacher student experiments and sparse function regression problems. In both cases the effect of the prior hyperparameter setting is evaluated and it is shown that the theoretically motivated value leads to best performance.

Strengths: The paper applies the insights derived from theoretical considerations well to empirical evaluation scenarios. The experiments are detailed and the suggested approach is very well evaluated on multiple toy and real world datasets. The claims and theoretical results of the paper are thus well supported by empirical evidence. Overall, I would consider the work significant and of relevance to the NeurIPS community.

Weaknesses: The paper introduces a theoretically sound way for the selection of the hyperparameter of the prior $\lambda$. This hyperparameter can to my understanding be interpreted as the prior expected amount of sparsity in the weights and thus could also be utilized when benchmarking the other methods. This should for example easily be possible for the variational dropout baseline and would allow to better understand if the higher performance is due to the more theoretically motivated selection of $\lambda$ or the way the spike and slap prior is implemented. Further, I cannot find a description over which hyperparameter range the sparsity was evaluated for the approaches which require a prior definition of the degree of sparsity (AGP and LOT). Here it would be relevant to know if the exact degree of sparsity was in the hyperparameter grid. Also, similar to the previous point, how do these models perform, when the prior degree of sparsity is set to $\lambda$? Finally, I would be interested in seeing a comparison to methods with "adaptive sparsity" such as the horseshoe prior. Here a prior estimate of the sparsity degree as derived in the theoretical results can also be included in the prior definition.

Correctness: To my understanding the claims and the methodology of the paper are correct. Yet, I am not an expert in this particular field.

Clarity: The paper is well written and clear.

Relation to Prior Work: While the paper refers to previous work, it is in my opinion too coarse in it's description. In particular, the statement "XYZ propose Bayesian Methods, but the choice of prior specification still needs to be determined.", does not clearly describe how the previous work is related besides proposing Bayesian Methods. Here, a more nuanced description would be beneficial.

Reproducibility: Yes

Additional Feedback: Minor mistakes: Missing word: line 156: , then *the* existing result Some references are out of date and refer to publications on arXiv while the papers have actually been published in proceedings (Ghosh and Doshi-Velez 2017). There might be a mistake in the equations after line 18 of the appendix. Here the sum seems to be iterating over the wrong variable. In the proof of Lemma 4.1 in the appendix, condition 4.4 seems to be required. This is in contrast to the main text where Lemma 4.1 is assumed to be valid under conditions 4.1-4.3. Questions: Regarding Table 1: At which level was a weight considered off / on. Or is this the average number of active weights for draws from the approximate posterior? ----- Read other reviews and author response. The authors address some of my questions but not all. In particular, it seems like the methods which require the definition of a degree of sparsity were not tested with the degree of sparsity set to the true sparsity of the data. Thus, they would never be able to give optimal performance. This is sensible, as the degree of sparsity is usually not known ahead of time, but should be explicitly mentioned to support the interpretation of the results. I encourage the authors to reflect this point in the final version of the paper.

[Author Response · NeurIPS 2020]

We thank the reviewers for their positive comments and constructive suggestions. We first address the key common
concerns and then move to individual reviewers. The paper will be updated accordingly in the camera-ready version.

**Comparison with prior works (R1, R2 and R4):** Besides the novel theoretical contribution, our work distinguished
itself from prior works on Bayesian sparse neural network by imposing a spike-and-slab prior with the Dirac spike
function. Hence automatically, all posterior samples are from exact sparse DNN models. In contrast, Blundell et al.
(ICML 2015) and Deng et al. (NIPS 2019) considered spike-and-slab priors with a Gaussian and Laplacian spike
respectively, while Ghosh et al. (ICML 2018) chose to use the popular horseshoe shrinkage prior. These existing works
actually yield posteriors over the dense DNN model space despite applying sparsity induced priors. In order to derive
explicit sparse inference results, user has to additionally determine certain pruning rules on the posterior, which is
similar to variational dropout (Molchanov et al. ICML 2017). Thus in our mind, they still belong to the category of
Bayesian pruning methods. However, we do agree with the reviewers that additional experiments could be made for
better comparison. For example, we conduct Horseshoe BNN under Simulation setting I: it achieves test RMSEs of
$1.02 \pm 0.01$ (A) and $1.24 \pm 0.03$ (B), with neuron-wise sparsity ratio of $0.79 \pm 0.19$ (A) and $0.81 \pm 0.09$ (B). This
performance is worse than our method. Note that more experiments will be added in the final version.

**Extend current result to more complicated network models (R1, R2 and R3):** Our developed theory should work
as long as it is a regular DNN (i.e., networks only involve fully-connected layers). Unfortunately we are unable to
perform super large-scale experiments due to limited computing resources at hand. Extending current results to more
complicated networks (convolutional layer, residual network, etc.) is not trivial. Conceptually, it requires design of
structured sparsity (e.g., group sparsity in Neklyudov et al. NIPS 2017) to serve the purpose of faster prediction.
Theoretically, it requires deeper understanding of the expressive ability (i.e. approximation error) and capacity (i.e.,
packing or covering number) of the network model space. By intuition, we conjecture that the generalized theory should
still hold, but it will be a future work to provide rigorous theoretical support. To illustrate the practical value of our
method for complex tasks, as a preliminary experiment, we apply a 2-Conv-2-FC network on Fashion-MNIST. The
testing accuracy is $90.07\%$ with $60\%$ sparsity (connection-wise), where the baseline by dense model is $90.65\%$.

**R2:** Please refer to common concerns for major questions, and below is our response to minor questions:
- To induce structured group sparsity, say in CNN modeling, we can let all weights in the same filter share the same
binary indicator, such that the whole filter is turned on/off simultaneously. The theoretical correctness of this approach
is left as our future work.
- The reviewer is correct. Molchanov et al. (ICML 2017) is a Bayesian pruning method, and the word "frequentist" on
Line 112/113 is a typo that should be removed.
- The five datasets are chosen for numerical study since they have fairly larger sample size.
- Throughout the paper, we always report the standard deviation. The "standard error" mentioned in the supplementary
material is a typo.

**R3:** Our results can be extended to different variation distribution families as long as they contain a Dirac component
at zero (although the technical details might differ), and the choice of normal slab distribution in this paper is merely for
technical simplicity. Without such a Dirac component, the variational posterior can no longer automatically induce
sparse inferences. Therefore, those cases are not under our consideration.

The authors don't possess cutting-edge infrastructure, thus reporting the CPU/GPU time of our implementation is
not very meaningful. Relatively, under same computational environment, our method takes roughly twice more time
than the frequentist counterparts, which is majorly due to the reason that there are more parameters to optimize in our
algorithm. Developing more computational efficient algorithm could be a future direction.

**R4:** The proposed choice of $\lambda$ does encourage posterior sparsity, but it doesn't play a dominating role on how sparse
the posterior would be. Instead, given the value of $\lambda$, it is the likelihood of the data that helps to adaptively choose the
data-dependent optimal posterior sparsity level. $\lambda$ plays as a regularization, and we provide theoretical suggestion such
that it doesn't mask the importance of the data likelihood. In contrast, pruning methods such as AGP and LOT require
user-input knowledge to explicitly determine the sparsity level of the result. In our implementation of AGP and LOT,
we tried grid search for the sparstiy levels ranging from 95% to 5%, and the level that yields the best testing accuracy is
chosen and reported.

For minor mistakes,
- Thanks for pointing out the typo and the outdated references. In Line 18, the sum should be iterated over $\gamma \in \Gamma^T$.
- Condition 4.4 is not required by Lemma 4.1 since we prove for cases either $\lim n(r_n^* + \xi_n^*) = \infty$ or $\lim n(r_n^* + \xi_n^*) \neq \infty$.
- For the question raised, as has been described in Line 214/215, the posterior sparsity is measured by $\sum_{i=1}^{T} \phi_i / T$ and
an individual connection (of the VB point estimator) is considered inactive if $\phi_i < 0.5$.

[Meta-Review · NeurIPS 2020]

The reviewers agree that this is interesting, rigorous, and novel work that explores sparsity in deep neural networks. While the assumptions required for consistency are not as general as one would hope, this paper lays the groundwork for new research directions. One weakness that the reviewers wish to see addressed is the lack of discussion of similar approaches in the BNN literature and how one might extend this approach to more complex models.